# DRB1, DRB2 and DRB4 Are Required for an Appropriate miRNA-Mediated Molecular Response to Osmotic Stress in *Arabidopsis thaliana*

**DOI:** 10.3390/ijms252312562

**Published:** 2024-11-22

**Authors:** Joseph L. Pegler, Jackson M. J. Oultram, Christopher P. L. Grof, Andrew L. Eamens

**Affiliations:** 1Centre for Plant Science, School of Environmental and Life Sciences, College of Engineering, Science and Environment, University of Newcastle, Callaghan, NSW 2308, Australia; joseph.pegler@newcastle.edu.au (J.L.P.); chris.grof@newcastle.edu.au (C.P.L.G.); 2School of Agriculture and Food Sustainability, The University of Queensland, St. Lucia, QLD 4072, Australia; 3Seaweed Research Group, School of Health, University of the Sunshine Coast, Maroochydore, QLD 4558, Australia

**Keywords:** *Arabidopsis thaliana* (*Arabidopsis*), osmotic stress, mannitol, microRNA (miRNA), miRNA-directed gene expression regulation, double-stranded RNA binding (DRB) protein, DRB1, DRB2, DRB4

## Abstract

*Arabidopsis thaliana* (*Arabidopsis*) double-stranded RNA binding (DRB) proteins DRB1, DRB2 and DRB4 perform essential roles in microRNA (miRNA) production, with many of the produced miRNAs mediating aspects of the molecular response of *Arabidopsis* to abiotic stress. Exposure of the *drb1*, *drb2* and *drb4* mutants to mannitol stress showed *drb2* to be the most sensitive to this form of osmotic stress. Profiling of the miRNA landscapes of mannitol-stressed *drb1*, *drb2* and *drb4* seedlings via small RNA sequencing, and comparison of these to the profile of mannitol-stressed wild-type *Arabidopsis* plants, revealed that the ability of the *drb1* and *drb2* mutants to mount an appropriate miRNA-mediated molecular response to mannitol stress was defective. RT-qPCR was next used to further characterize seven miRNA/target gene expression modules, with this analysis identifying DRB1 as the primary DRB protein required for miR160, miR164, miR167 and miR396 production. In addition, via its antagonism of DRB1 function, DRB2 was shown by RT-qPCR to play a secondary role in regulating the production of these four miRNAs. This analysis further showed that DRB1, DRB2 and DRB4 are all required to regulate the production of miR399 and miR408, and that DRB4 is the primary DRB protein required to produce the non-conserved miRNA, miR858. Finally, RT-qPCR was used to reveal that each of the seven characterized miRNA/target gene expression modules responded differently to mannitol-induced osmotic stress in each of the four assessed *Arabidopsis* lines. In summary, this research has identified mannitol-stress-responsive miRNA/target gene expression modules that can be molecularly manipulated in the future to generate novel *Arabidopsis* lines with increased tolerance to this form of osmotic stress.

## 1. Introduction

In plants, microRNAs (miRNAs) function as essential regulatory molecules to modulate the expression of genes central to the molecular response of a plant to a wide range of abiotic stresses [1,2,3,4]. In the genetic model plant *Arabidopsis thaliana* (*Arabidopsis*), the accumulation level of specific miRNAs, and therefore the altered expression of their target genes, has been described post its exposure to osmotic stress [5,6,7]. For example, Ma et al. [8] showed that in response to exposure to the osmotic stress agent polyethylene glycol (PEG), miR408 abundance was reduced and the expression of its target genes, including *PLANTACYNIN, CUPREDOXIN, UCLACYANIN* and *LACCASE3*, was elevated in two-week-old *Arabidopsis* seedlings. Furthermore, the authors went on to show that compared to wild-type plants, *Arabidopsis* lines molecularly modified to overexpress miR408 were more sensitive to PEG-induced osmotic stress, whereas an *Arabidopsis* mutant line where miR408 production is defective was more resistant to PEG-induced osmotic stress [8]. Therefore, considering the findings detailed in such studies [5,6,7,8], it is unsurprising that the focus of contemporary research is aimed at constructing a more detailed understanding of the fundamental miRNA-mediated molecular responses of *Arabidopsis* to osmotic stress.

Most plant miRNAs are processed from non-coding RNAs (ncRNAs) transcribed from individual transcription units that contain their own promoter and terminator regions, which (1) allows for the spatiotemporal accumulation of the miRNA post processing from its ncRNA precursor, and (2) directs *MICRORNA* (*MIR*) gene transcription by RNA polymerase II, the same RNA polymerase responsible for protein-coding gene expression [9,10]. Due to a region of partial sequence self-complementarity, the ncRNA folds to form a stem-loop structured double-stranded RNA (dsRNA), termed the primary-miRNA (pri-miRNA). The pri-miRNA is bound by SERRATE1 (SE1), which transports the precursor transcript to Dicing bodies (D-bodies) in the nucleus of the plant cell where it is processed by the Dicing complex (D-complex) [11,12,13]. The D-complex contains DICER-LIKE1 (DCL1) and HYPONASTIC LEAVES1 (HYL1, also called dsRNA BINDING1 (DRB1)) at its functional core with DRB1 accurately positioning DCL1 on the pri-miRNA to ensure efficient DCL1-catalyzed processing of the ncRNA to generate the precursor-miRNA (pre-miRNA) [12,13,14,15,16]. The D-complex proteins, DCL1 and DRB1, continue to function together to further process specific structural features of the pre-miRNA to produce the miRNA/miRNA* duplex, with the 3′ terminal nucleotide of both duplex strands methylated by the small RNA (sRNA)-specific methyltransferase, HUA ENHANCER1 (HEN1) [17]. The two duplex strands are subsequently separated from one another via cleavage of the miRNA* passenger strand and retention of the miRNA guide strand by ARGONAUTE1 (AGO1) [18,19]. miRNA-loaded AGO1 forms the catalytic core of the miRNA-induced silencing complex (miRISC), with the miRNA guide strand used by miRISC as a sequence-specificity determinant to direct expression repression of messenger RNA (mRNA) transcripts that harbor highly complementary target sequences via an AGO1-catalyzed target transcript cleavage mode of RNA silencing [19,20].

In addition to DRB1, *Arabidopsis* encodes four other DRB proteins, DRB2 to DRB5, with the two other nucleus-localized DRB proteins, DRB2 and DRB4, also assigned functional roles in the production of specific miRNA subsets [21,22,23,24,25,26,27,28]. In an analogous fashion to the DCL1/DRB1 partnership for the production of the conserved miRNA subclass, DRB4 assists DCL4 to process the non-conserved miRNA subclass from their dsRNA precursors [23,24,25]. However, in contrast to the widespread requirement of DCL1 and DRB1 for conserved miRNA production, the DCL4/DRB4 partnership plays a much more restricted role in miRNA production in *Arabidopsis*, only producing a small subset of non-conserved miRNAs, which are processed from dsRNA precursors with high levels of complementarity. That is, precursor transcripts with comprehensive base pairing between the two stem arms of the pri-miRNA/pre-miRNA, which are separated from each other by a short loop region [22,23]. Likely via its ability to form functional interactions with either DCL1 or DCL4, DRB2 is also involved in the production of specific subsets of both the conserved and non-conserved subclasses of miRNA in *Arabidopsis* tissues, but only in the tissues where the expression of *DRB2* overlaps with the expression domains of either *DRB1* or *DRB4* [22,23,26,28]. Furthermore, DRB1 and DRB2 have been proposed to also function as molecular switches to determine if a specific miRNA directs the canonical mode of miRNA-directed RNA silencing in plants, target mRNA cleavage, or if the DRB2-dependent miRNA regulates the expression of its target genes via the alternate mode of miRNA-directed RNA silencing in plants, translational repression [26,27,28,29].

Considering that in *Arabidopsis* [5,6,7,8], and in other plant species including potato (*Solanum tuberosum*), wheat (*Triticum aestivum*) and soybean (*Glycine max*) [30,31,32], miRNA/target gene expression modules have been demonstrated to form part of the molecular response to osmotic stress, and in *Arabidopsis*, DRB1, DRB2 and DRB4 are required for miRNA production [14,15,16,21,22,23,24,25,26,27,28,29], the phenotypic and molecular response of the *Arabidopsis* knockout mutants *drb1*, *drb2* and *drb4* was assessed post their exposure to the osmotic stress agent mannitol. More specifically, 8-day-old wild-type *Arabidopsis* (Columbia-0 (Col-0)) seedlings and the *drb1*, *drb2* and *drb4* single mutants were exposed to a 7-day cultivation period in the presence of 200 millimolar (200 mM) mannitol to document the miRNA-mediated molecular responses of these four *Arabidopsis* lines to this form of osmotic stress. When the phenotypic and physiological analyses performed in this study were considered together, and via comparison of *drb* mutant characteristics to mannitol-stressed Col-0 seedlings, the *drb2* mutant appeared to be the most sensitive to the imposed stress. Profiling by small RNA sequencing (sRNA-Seq) of the mannitol stress responsive miRNA landscapes of Col-0, *drb1*, *drb2* and *drb4* seedlings, revealed considerable differences in the global miRNA-mediated response to the imposed stress. Specifically, in mannitol-stressed Col-0 and *drb4* seedlings, 96.7% and 90.5% of miRNAs with altered abundance were increased in their levels, whereas in direct contrast, 94.0% and 70.7% of miRNAs with altered levels in mannitol-stressed *drb1* and *drb2* seedlings were reduced in their abundance. These findings showed that the miRNA-mediated molecular response to mannitol-induced osmotic stress was defective in the absence of the functional activity of DRB1 and DRB2. We also found that of the seven miRNAs selected for further molecular characterization, DRB1 acted as the primary DRB protein family member required for miR160, miR164, miR167 and miR396 production. However, via its antagonism of DRB1 function, DRB2 was revealed to mediate a secondary role in regulating the production of these four miRNAs in *Arabidopsis* seedlings. DRB1 was also identified as the primary DRB protein required for miR399 production, but that both DRB2 and DRB4 function synergistically to aid the action of DRB1 to fine-tuning the rate of production of miR399 in *Arabidopsis* seedlings. In addition, the hierarchical action of DRB1, DRB2 and DRB4 was revealed to be required for miR408 production, and DRB4 was the primary DRB protein family member required to produce the non-conserved miRNA, miR858. Finally, we show that the expression modules of the seven selected miRNAs responded differently to mannitol-induced osmotic stress in each of the four *Arabidopsis* lines assessed in this study. In summary, this research successfully identified specific miRNA/target gene expression modules responsive to mannitol stress that can be targeted for molecular manipulation in the future to generate novel *Arabidopsis* lines with increased tolerance to this form of osmotic stress.

## 2. Results

### 2.1. Phenotypic and Physiological Assessment of 15-Day-Old Col-0, drb1, drb2 and drb4 Seedlings Following a 7-Day Osmotic Stress Treatment Period

Under standard *Arabidopsis* growth conditions, the differences in the developmental progression of the *drb1*, *drb2* and *drb4* single knockout mutants compared to Col-0 plants of the same age have been described in detail previously [23,24,26]. In brief, compared to 15-day-old control-grown Col-0 (Col-0/Ns) seedlings, non-stressed *drb1* (*drb1*/Ns) seedlings display impeded development, as evidenced by the formation of narrow and elongated rosette leaves with hyponasty and a delay in the emergence of rosette leaf pairs (Figure 1A). In contrast, non-stressed *drb2* (*drb2*/Ns) seedlings exhibit promoted vegetative development, producing rosettes that are slightly larger than those of Col-0/Ns seedlings, primarily due to the formation of larger ovoid rosette leaves on elongated petioles (Figure 1A). Figure 1A also shows the penetrance phenotype associated with the loss-of-function DRB2 mutation in *Arabidopsis*, with a small percentage (~10–15%) of *drb2*/Ns plants exhibiting highly promoted vigor compared with the more subtle enhancement of vegetative development displayed by most other individuals within the same population. At 15 days of age, the rosettes of non-stressed *drb4* (*drb4*/Ns) seedlings are mildly reduced in size compared to those of Col-0/Ns seedlings, largely due to the development of rosette leaves with epinasty (Figure 1A). Figure 1A also clearly shows the severe restriction of the development of 15-day-old Col-0, *drb1*, *drb2* and *drb4* seedlings following the 7-day cultivation period on solid growth medium supplemented with 200 mM mannitol, an osmotic stress agent. More specifically, the rate of rosette leaf emergence was not effected in mannitol-stressed Col-0, *drb1*, *drb2* and *drb4* seedlings (referred to herein as Col-0/Mann, *drb1*/Mann, *drb2*/Mann and *drb4*/Mann seedlings); however, for all rosette leaves which did emerge, each leaf was of a much smaller size, formed on a petiole of reduced length, and these leaves were turned downwards (epinasty) at their distal tips towards the growth medium (Figure 1A).

Figure 1B,C readily show the extent of development restriction of mannitol-stressed Col-0, *drb1*, *drb2* and *drb4* seedlings when compared to the control-grown counterpart of each *Arabidopsis* line. More specifically, quantification of the fresh weight of whole seedlings revealed that compared to the Col-0/Ns, *drb1*/Ns, *drb2*/Ns and *drb4*/Ns samples, the whole-seedling fresh weight of Col-0/Mann, *drb1*/Mann, *drb2*/Mann and *drb4*/Mann seedlings was significantly reduced by 68.4%, 64.5%, 54.9% and 57.6%, respectively (Figure 1B). Similarly, quantification of total rosette area showed significant reductions of 61.7%, 59.4%, 74.7% and 59.6% for Col-0/Mann, *drb1*/Mann, *drb2*/Mann and *drb4*/Mann seedlings when compared to the respective control-grown counterpart of each line (Figure 1C). Similar to the phenotypic metrics of whole-seedling fresh weight and total rosette area, quantification of primary root length provided further evidence of the negative impact the 7-day mannitol stress period had on the development of *drb1*/Mann and *drb2*/Mann seedlings, with the primary root length of mannitol-stressed *drb1* and *drb2* seedlings significantly reduced by 29.0% and 17.3%, respectively (Figure 1D). Interestingly, primary root length was only mildly decreased by 7.3% in the *drb4*/Mann sample and was mildly promoted by 4.8% in Col-0/Mann seedlings (Figure 1D). However, except for the mild increase to the primary root length of Col-0/Mann seedlings, quantification of the phenotypic metrics of whole-seedling fresh weight, rosette area and primary root length, all readily showed the severe negative impact the 7-day growth period in the presence of 200 mM mannitol had on the development of 15-day-old Col-0, *drb1*, *drb2* and *drb4* seedlings.

The total content of the flavonoid pigment, anthocyanin, and the key photosynthetic pigments, chlorophyll *a* and *b* (Chl *a* and Chl *b*), were next quantified in mannitol-stressed Col-0, *drb1*, *drb2* and *drb4* seedlings for comparison to the control-grown counterpart of each assessed *Arabidopsis* line. As can be observed in Figure 1A, quantification of anthocyanin content showed that the 7-day mannitol stress period did not influence the accumulation of this flavonoid pigment to any significant degree in the Col-0/Mann, *drb1*/Mann, *drb2*/Mann or *drb4*/Mann samples (Figure 1E). Moreover, anthocyanin content was mildly reduced by 0.5% and 3.5% in Col-0/Mann and *drb2*/Mann seedlings, and moderately elevated by 23.6% and 8.6% in the mannitol-stressed *drb1* and *drb4* samples, respectively (Figure 1E). In contrast to this finding, the content of Chl *a* and *b* was significantly reduced in Col-0/Mann, *drb1*/Mann and *drb4*/Mann seedlings, with Chl *a* content decreased by 23.8%, 34.9% and 32.0% (Figure 1F) and Chl *b* levels reduced by 23.4%, 24.5% and 31.1%, respectively (Figure 1G). The content of the two assessed photosynthetic pigments was reduced by a much greater degree in the *drb2*/Mann sample, with Chl *a* levels reduced by 57.7% and the content of Chl *b* decreased by 58.1% (Figure 1F,G). In summary, three of the six assessed phenotypic and physiological metrics, including rosette area, Chl *a* and Chl *b* content, showed the greatest extent of negative impact in the *drb2*/Mann sample, indicating that the *drb2* mutant was the most sensitive to the imposed stress.

### 2.2. Molecular Profiling of the miRNA Pathway Genes DCL1, DRB1, DRB2 and DRB4 in 15-Day-Old Col-0, drb1, drb2 and drb4 Seedlings Following Mannitol-Induced Osmotic Stress

In *Arabidopsis*, the *Δ1-PYRROLINE-5-CARBOXYLATE SYNTHETASE1* (*P5CS1*) gene (*AT2G39800*) forms a well-characterized stress response gene [33,34]. Therefore, to demonstrate that the 7-day growth period on solid *Arabidopsis* growth medium supplemented with 200 mM mannitol had invoked a stress response at the molecular level in 15-day-old Col-0, *drb1*, *drb2* and *drb4* seedlings, reverse transcriptase quantitative PCR (RT-qPCR) was used to assess the expression of the *P5CS1* gene. Figure 2A shows that the applied stress regime had indeed induced a stress response at the molecular level in all four of the assessed *Arabidopsis* lines, with RT-qPCR revealing *P5CS1* transcript abundance to be significantly elevated by 1.7-, 8.0-, 2.4- and 2.9-fold in Col-0/Mann, *drb1*/Mann, *drb2*/Mann and *drb4*/Mann seedlings compared to the respective level of *P5CS1* expression in Col-0/Ns, *drb1*/Ns, *drb2*/Ns and *drb4*/Ns seedlings. Interestingly, this initial molecular analysis further revealed that the degree of *P5CS1* expression induction was greater in all three of the assessed *drb* single mutants than it was in the Col-0/Mann sample (Figure 2A), a finding which putatively suggested that the ability to mount an appropriate molecular response to this form of osmotic stress was compromised in each *drb* mutant.

The requirement of the core functional DCL1/DRB1 partnership to produce most *Arabidopsis* miRNAs is well documented [14,15,16]. Similarly, the requirement of DRB2, most likely via formation of a functional partnership with DCL1, and DRB4 together with DCL4 to produce specific miRNA cohorts in *Arabidopsis*, is also well established [22,23,26,27,28]. Therefore, RT-qPCR was next applied to reveal any change in *DCL1*, *DRB1*, *DRB2* and *DRB4* gene expression in Col-0, *drb1*, *drb2* and *drb4* plants that may have been induced by the 7-day cultivation period in the presence of 200 mM mannitol. Compared to non-stressed Col-0 seedlings, RT-qPCR showed *DCL1* and *DRB1* gene expression to be decreased by 1.7-fold (Figure 2B,C), and that the level of expression of the *DRB2* and *DRB4* genes was mildly elevated by 1.3- and 1.1-fold, respectively, in Col-0/Mann seedlings (Figure 2D,E). In *drb1*/Mann seedlings, *DCL1* and *DRB4* gene expression was significantly increased by 1.7- and 2.9-fold, respectively (Figure 2B,E). However, RT-qPCR also revealed that the abundance of the *DRB2* transcript remained unchanged between *drb1*/Ns and *drb1*/Mann seedlings (Figure 2D). In *drb2*/Mann seedlings, *DCL1* and *DRB1* gene expression was mildly reduced by 1.2- and 1.1-fold respectively (Figure 2B,C) and *DRB4* expression was significantly reduced by 2.4-fold (Figure 2E). Similarly, in *drb4*/Mann seedlings, *DCL1* and *DRB2* gene expression was mildly and significantly decreased by 1.3- and 2.6-fold, respectively (Figure 2B,D). In contrast, *DRB1* transcript abundance was significantly elevated by 2.9-fold in mannitol-stressed *drb4* seedlings (Figure 2C).

### 2.3. Profiling of the miRNA Landscapes of 15-Day-Old Col-0, drb1, drb2 and drb4 Seedlings Post a 7-Day Exposure Period to the Osmotic Stress Agent Mannitol by Small RNA Sequencing

With RT-qPCR revealing *DCL1*, *DRB1*, *DRB2* and *DRB4* expression to be altered differently in Col-0/Mann, *drb1*/Mann, *drb2*/Mann and *drb4*/Mann seedlings by the imposed stress (Figure 2B–E), sRNA-Seq was next applied to determine what effect altered expression of the genes that encode these four core pieces of protein machinery of the production stage of the miRNA pathway had on the global miRNA populations of mannitol-stressed Col-0, *drb1*, *drb2* and *drb4* seedlings. Sequencing of the miRNA populations of control-grown Col-0, *drb1*, *drb2* and *drb4* seedlings initially identified 262, 221, 258 and 295 miRNAs, respectively. Figure 3 shows that the abundance of many of the miRNAs detected in Col-0, *drb1*, *drb2* and *drb4* seedlings was altered by the 7-day growth period on solid *Arabidopsis* medium supplemented with 200 mM mannitol. Specifically, compared to the miRNA profiles established for Col-0/Ns, *drb1*/Ns, *drb2*/Ns and *drb4*/Ns seedlings, sRNA-Seq revealed that the abundance of 123, 100, 58 and 95 miRNAs was altered (either elevated or reduced) by a significant degree in Col-0/Mann, *drb1*/Mann, *drb2*/Mann and *drb4*/Mann seedlings, respectively. This showed that of the total number of miRNAs detected in non-stressed Col-0, *drb1*, *drb2* and *drb4* seedlings, the applied mannitol stress significantly altered the abundance of 46.9%, 45.2%, 22.5% and 32.2% of the global miRNA populations of the four assessed *Arabidopsis* lines. It is important to note here that the higher degree of abundance alteration in mannitol-stressed Col-0 and *drb1* seedlings, compared with that determined for *drb2*/Mann and *drb4*/Mann seedlings, may simply reflect the more restricted involvement of DRB2 and DRB4 in the production stage of the *Arabidopsis* miRNA pathway [22,23,26,27]. Alternatively, this result could indicate that the loss of DRB2 or DRB4 function negatively impacted the ability of *Arabidopsis* to mount an effective miRNA-directed molecular response to this form of osmotic stress to a greater degree than did the loss of DRB1 function.

The profiling of miRNAs with significantly altered abundance in Col-0/Mann, *drb1*/Mann, *drb2*/Mann and *drb4*/Mann seedlings further shows that the imposed 7-day mannitol stress period had a different influence on the global miRNA populations of Col-0 and *drb4* seedlings than it did on the miRNA populations of *drb1*/Mann and *drb2*/Mann seedlings (Figure 3). Moreover, in Col-0/Mann seedlings, a highly pronounced trend of promoted miRNA accumulation was observed, with 119 of the 123 (96.7%) significantly altered miRNAs increased in abundance in Col-0/Mann plants. Similarly, in *drb4*/Mann seedlings, 86 of the 95 miRNAs (90.5%) with significantly altered abundance post exposure to the applied stress were increased in their abundance. In contrast, Figure 3 also shows the general downward trend in miRNA accumulation in the mannitol-stressed *drb1* and *drb2* samples when compared to their respective control-grown counterparts. This downward trend in miRNA abundance was much more pronounced in the *drb1* mutant than it was in the *drb2* mutant post exposure to mannitol stress, with 94.0% (n = 94/100) of miRNAs with significantly altered abundance having a reduced level of accumulation in the *drb1*/Mann sample. In comparison, 70.7% (n = 41/58) of the miRNAs with significantly altered abundance in *drb2*/Mann seedlings were reduced in their level of accumulation. When considered together, the opposing trends in miRNA abundance alteration in mannitol-stressed Col-0 and *drb4* seedlings, compared to *drb1*/Mann and *drb2*/Mann seedlings, readily demonstrates that the miRNA-mediated molecular response to this form of osmotic stress differed considerably in these two *drb* mutant backgrounds compared to that documented for Col-0 or *drb4* seedlings.

Small RNA sequencing clearly showed that mannitol-induced osmotic stress altered the abundance of many of the miRNAs that accumulate in 15-day-old *Arabidopsis* whole seedlings (Figure 3). RT-qPCR was next applied to quantify the accumulation trends of members of seven *MICRORNA* (*MIR*) gene families, including the *MIR160*, *MIR164*, *MIR167*, *MIR396*, *MIR399*, *MIR408* and *MIR858* gene families. Appendix A shows that the abundance of all members of the *MIR160*, *MIR164*, *MIR167*, *MIR396* and *MIR399* gene families was reduced in control-grown *drb1* seedlings (Appendix A). This formed an expected result, considering that DRB1 is the preferred functional partner of DCL1 for the production of almost all miRNAs that accumulate in *Arabidopsis* tissues [14,15,16,21]. In contrast, sRNA-Seq showed that the abundance of miR408, miR858a and miR858b was elevated in non-stressed *drb1* whole seedlings (Appendix A). The abundance of miR408 was also increased in non-stressed *drb2* and *drb4* seedlings. This forms an interesting result and suggests that the functional activity of all three nucleus-localized DRB proteins are required to regulate the processing of this miRNA from its precursor transcript, *PRE-MIR408*, in *Arabidopsis* whole seedlings (Appendix A). For the two members of the *MIR858* gene family detected by sRNA-Seq, miR858a abundance remained unchanged in the *drb2*/Ns sample and was reduced in the *drb4*/Ns sample. Furthermore, the level of miR858b was elevated in the *drb2*/Ns sample and was reduced in the *drb4*/Ns sample, which, when considered together with the *drb1*/Ns sequencing data (Appendix A), identified DRB4 as the likely DRB protein required for *MIR858* gene family member production in *Arabidopsis*.

The majority of members of the *MIR160*, *MIR164*, *MIR167* and *MIR396* gene families were increased in abundance in non-stressed *drb2* seedlings (Appendix A), and considering that DRB2 is antagonistic towards the action of DRB1 in the DCL1/DRB1 functional partnership [26,27], this finding further identified the requirement of DRB2 to add an additional layer of regulatory complexity to the production of members of these four *MIR* gene families in *Arabidopsis* (Appendix A). Interestingly, the abundance of five (miR399a, miR399b, miR399c, miR399e and miR399f) of the six members of the *MIR399* gene family was revealed by sRNA-Seq to also be reduced in the *drb2*/Ns and *drb4*/Ns samples, in addition to the *drb1*/Ns sample (Appendix A). This finding suggested that DRB1, DRB2 and DRB4 all contribute to *MIR399* gene family member production in *Arabidopsis* tissues. The general abundance trend of reduced family member accumulation documented by sRNA-Seq in the *drb1*/Ns sample was closely mirrored in *drb4*/Ns seedlings, with the level of abundance of each member of the *MIR160*, *MIR164*, *MIR167* and *MIR396* gene families reduced in non-stressed *drb4* whole seedlings (Appendix A). This formed an unexpected result, as DRB4 has been previously demonstrated to specifically form a functional partnership with DCL4 to produce the non-conserved subclass of *Arabidopsis* miRNAs [22,23]. However, the *MIR160*, *MIR164*, *MIR167* and *MIR396* gene families are all highly conserved miRNAs in plants, and therefore, reduced miR160, miR164, miR167 and miR396 abundance in the *drb4*/Ns sample indicated that in addition to DRB1, and to a lesser degree DRB2, DRB4 also mediates a role in fine-tuning the rate of production of the individual members of these four *MIR* gene families.

Summing together the levels of accumulation of all family members of the *MIR160*, *MIR164*, *MIR167*, *MIR396*, *MIR399*, *MIR408* and *MIR858* gene families in non-stressed *drb1*, *drb2* and *drb4* seedlings for comparison to the Col-0/Ns sample (Appendix A) confirmed that DRB1 is the primary DRB protein required for production of members of the *MIR160*, *MIR164*, *MIR167*, *MIR396* and *MIR399* gene families, and further, that DRB1 appears to mediate a secondary role in miR408 and miR858 production (Appendix A). This analysis also showed the contribution of DRB2 to regulate the production of members of the *MIR160*, *MIR164*, *MIR167* and *MIR396* gene families via antagonism of the DRB1/DCL1 functional partnership [26,27], and likely via antagonism of the DCL4/DRB4 functional partnership [23] for *MIR858* family member production (Appendix A). Appendix A also shows that in addition to DRB1, DRB2 (likely via its interaction with DCL1) plays a secondary role for *MIR399* family member production. Considering that the DCL4/DRB4 functional partnership is well established as the primary DCL/DRB partnership required for the production of the non-conserved subclass of miRNAs in *Arabidopsis* [22,23], it was somewhat unexpected to observe an overall decrease to the level of abundance of the *MIR160*, *MIR164*, *MIR167*, *MIR396* and *MIR399* gene families in *drb4*/Ns seedlings (Appendix A), a finding which putatively identified a secondary role for DRB4 to that of DRB1 for the production of members of these five *MIR* gene families. In addition, based on the total abundance of both members of the *MIR858* gene family (Appendix A), this analysis readily identified DRB4 as the primary DRB protein required for miR858 production from its precursor transcripts, *PRE-MIR858A* and *PRI-MIR858B*. In summary, the unique *MIR* gene family member accumulation profiles specific to each *drb* mutant background, constructed via the analysis of the sRNA-Seq data obtained for control-grown Col-0, *drb1*, *drb2* and *drb4* seedlings, readily identified the requirement to further analyze these seven *MIR* gene families by RT-qPCR in mannitol-stressed Col-0, *drb1*, *drb2* and *drb4* seedlings.

### 2.4. Quantification of miRNA Abundance via RT-qPCR in 15-Day-Old Col-0, drb1, drb2 and drb4 Seedlings Following a 7-Day Period of Exposure to the Osmotic Stress Agent Mannitol

Profiling of the abundance of the three *MIR160* gene family members by sRNA-Seq in control-grown *drb1*, *drb2* and *drb4* seedlings showed that the abundance of miR160a, miR160b and miR160c was decreased in *drb1*/Ns and *drb4*/Ns seedlings and elevated in *drb2*/Ns seedlings, compared to its level of accumulation in Col-0/Ns seedlings (Appendix A), a finding that identified miR160 as a target for further analysis in mannitol-stressed *Arabidopsis* seedlings. Moreover, sRNA-Seq revealed that the summed abundance of members of the *MIR160* gene family was elevated in Col-0/Mann, *drb1*/Mann, *drb2*/Mann and *drb4*/Mann seedlings by the imposed stress (Figure 4A). RT-qPCR also showed that miR160 abundance was altered in all four *Arabidopsis* lines assessed following the 7-day cultivation period on standard *Arabidopsis* growth medium supplemented with 200 mM mannitol. However, in direct contrast to the sRNA-Seq data, RT-qPCR assessment of miR160 levels revealed that the accumulation of the quantified miRNA was reduced by 1.2-, 4.0-, 2.5- and 3.6-fold in Col-0/Mann, *drb1*/Mann, *drb2*/Mann and *drb4*/Mann seedlings, respectively (Figure 4B).

In rice (*Oryza sativa*), wheat (*Triticum aestivum*), potato (*Solanum tuberosum*) and desert poplar (*Populus euphratica*), miR164 has been shown to be responsive to mannitol or to other osmotic stress agents [35,36,37,38]. The abundance of miR164 was revealed by sRNA-Seq to be moderately elevated by mannitol stress in Col-0, *drb1*, *drb2* and *drb4* seedlings (Figure 4A). However, in contrast, RT-qPCR indicated that miR164 abundance was only mildly reduced by 1.1-fold in Col-0/Mann and *drb1*/Mann seedlings and by 1.2-fold in *drb2*/Mann and *drb4*/Mann seedlings (Figure 4C). Like miR164, miR167 has been previously reported to be responsive to osmotic stress in rice, barrel clover (*Medicago truncatula*) and *Arabidopsis* [5,35,39,40,41]. Both the sRNA-Seq and RT-qPCR analyses applied in this study confirmed that in *Arabidopsis* seedlings, miR167 is responsive to osmotic stress (Figure 4A,D). More specifically, RT-qPCR showed that the level of miR167 was moderately elevated by 1.8-fold in Col-0/Mann seedlings, significantly elevated by 2.1- and 2.0-fold in the *drb1*/Mann and *drb2*/Mann samples, and significantly reduced by 2.0-fold in mannitol-stressed *drb4* seedlings (Figure 4D).

The responsiveness of miR396 to osmotic stress has previously been shown in rice and barrel clover [35,39] and furthermore, in *Arabidopsis*, the well-established target gene of miR396-directed expression regulation, *GROWTH REGULATING FACTOR7* (*GRF7*), has been proposed to play a central role in mounting an adaptive response to mannitol stress [42]. Small RNA sequencing and RT-qPCR both revealed that in response to the imposed stress, miR396 abundance was significantly elevated in the Col-0/Mann (3.1-fold) and *drb4*/Mann (2.5-fold) samples, moderately elevated (1.5-fold) in the *drb2*/Mann sample and significantly reduced (−4.2-fold) in mannitol-stressed *drb1* seedlings (Figure 4A,E). The miR399/*PHOSPHATE2* (*PHO2*) expression module is essential for *Arabidopsis* to mount an adaptive response to conditions of low inorganic phosphorous (Pi) [43,44,45]. Furthermore, via a molecular modification approach, we have also shown that the miR399/*PHO2* expression module plays an important role in the molecular response of *Arabidopsis* to salt stress [46]. RT-qPCR showed that miR399 abundance was significantly elevated by 2.7-fold in Col-0/Mann seedlings, moderately elevated by 1.6- and 1.7-fold in *drb2*/Mann and *drb4*/Mann seedlings, and mildly reduced in its abundance by 1.3-fold in the *drb1*/Mann sample (Figure 4F), miRNA abundance trends which were also identified via sRNA-Seq (Figure 4A).

Previous research in *Arabidopsis* has shown that the insertion knockout mutant *mir408* is more resistant to the osmotic stress agent polyethylene glycol (PEG; 20% *w/v*) than are wild-type *Arabidopsis* seedlings, whereas miR408 overexpression lines are more sensitive to growth in an environment containing this osmotic stress agent [8]. Figure 4A shows that in response to the 7-day growth period in the presence of mannitol, miR408 was increased in abundance in Col-0/Mann and *drb1*/Mann seedlings and decreased in its level of accumulation in *drb2*/Mann and *drb4*/Mann seedlings. These sRNA-Seq identified abundance trends were confirmed via RT-qPCR, which showed that miR408 levels were significantly enhanced by 3.0- and 1.9-fold in Col-0/Mann and *drb1*/Mann seedlings and mildly reduced by 1.1- and 1.4-fold in the *drb2*/Mann and *drb4*/Mann samples, respectively (Figure 4G). To date, miR858 has only been identified as an osmotic stress responsive miRNA in *Ammopiptanthus nanus*, an evergreen broadleaf shrub of the northwest desert region of China [47]. Small RNA sequencing indicated that miR858 abundance was elevated in mannitol-stressed Col-0, *drb2* and *drb4* seedlings and reduced in *drb1*/Mann seedlings (Figure 4A). RT-qPCR confirmed that miR858 abundance was significantly reduced by 3.3-fold in the *drb1*/Mann sample; however, this analysis further indicated that the abundance of the miR858 sRNA was also significantly reduced by 10.0-, 3.8- and 2.7-fold in Col-0/Mann, *drb2*/Mann and *drb4*/Mann seedlings, respectively (Figure 4H). Together, these findings readily demonstrate that in *Arabidopsis* whole seedlings, miR858 is highly responsive to this form of osmotic stress.

### 2.5. Assessment of Target Gene Expression via RT-qPCR Analysis for miRNAs Demonstrated to Be Responsive to Osmotic Stress in 15-Day-Old Col-0, drb1, drb2 and drb4 Whole Seedlings

To correlate any change to the level of expression of a select target gene for each of the seven miRNAs shown to have altered abundance following mannitol stress, RT-qPCR was used. In *Arabidopsis*, miR160 regulates the expression of *AUXIN RESPONSE FACTOR10* (*ARF10*), *ARF16* and *ARF17*, three closely related members of the *ARF* gene family of transcription factors [48,49]. The level of *ARF17* expression remained unchanged in Col-0/Mann seedlings, was mildly decreased by 1.5- and 1.3-fold in *drb1*/Mann and *drb2*/Mann seedlings, and was moderately elevated by 1.8-fold in the *drb4*/Mann sample (Figure 5A). Considering that miR160 has been shown to regulate the abundance of *ARF17* via a transcript cleavage mode of RNA silencing [48], and that miR160 levels were reduced in all four assessed *Arabidopsis* lines following mannitol stress (Figure 4B), only the elevated *ARF17* gene expression in *drb4*/Mann seedlings (Figure 5A) formed an expected result. *CUP SHAPED COTYLEDON1* (*CUC1*) and *CUC2* belong to a small clade of the *NAC*-domain gene superfamily in *Arabidopsis,* whose expression is regulated at the posttranscriptional level by miR164 [50,51]. The level of *CUC1* transcript abundance was revealed by RT-qPCR to be significantly reduced by 2.0-, 2.9- and 3.0-fold in mannitol-stressed Col-0, *drb1* and *drb2* seedlings, respectively, and to be mildly reduced by 1.1-fold in the *drb4*/Mann sample (Figure 5B). Although RT-qPCR showed miR164 abundance to be mildly reduced in Col-0/Mann, *drb1*/Mann, *drb2*/Mann and *drb4*/Mann seedlings (Figure 4C), sRNA-Seq indicated that the abundance of the miR164 sRNA was elevated in the four assessed *Arabidopsis* lines (Figure 4A), a finding that may account for the reduced level of *CUC1* expression in mannitol-stressed Col-0, *drb1*, *drb2* and *drb4* plants.

Via controlling the pattern of expression of its *ARF* target genes, specifically *ARF6* and *ARF8*, miR167 has been previously shown to regulate diverse aspects of *Arabidopsis* growth and development, including regulating flower formation and root development [52,53]. *ARF8* expression was shown by RT-qPCR to be significantly reduced by 2.5-, 3.8- and 2.5-fold in the Col-0/Mann, *drb1*/Mann and *drb2*/Mann samples, respectively, and to be significantly elevated by 2.2-fold in *drb4*/Mann seedlings (Figure 5C). The targeting miRNA, miR167, showed an opposing abundance trend in all four assessed *Arabidopsis* lines following the 7-day cultivation period on *Arabidopsis* growth medium supplemented with 200 mM mannitol (Figure 4D). The miRNA and target gene expression profiles strongly indicated that miR167 controls *ARF8* expression via the canonical mode of miRNA-directed RNA silencing in plants, transcript cleavage, in *Arabidopsis* whole seedlings during osmotic stress. As stated above, Kim et al. [42] previously proposed that GRF7 directs a central role in the adaptive response of *Arabidopsis* to mannitol-induced osmotic stress. Figure 5D shows that compared to the control-grown counterpart of each *Arabidopsis* line, the expression of *GRF7* was reduced by 2.7-, 1.8-, 2.0- and 1.7-fold in Col-0/Mann, *drb1*/Mann, *drb2*/Mann and *drb4*/Mann seedlings, respectively. A decreased level of *GRF7* expression was expected in mannitol-stressed Col-0, *drb2* and *drb4* seedlings, considering that the abundance of the targeting miRNA, miR396, was increased by 3.1-, 1.5- and 2.5-fold in these three samples following the application of mannitol-induced osmotic stress (Figure 4E). In contrast, sRNA-Seq and RT-qPCR both showed that the level of accumulation of miR396 was reduced in *drb1*/Mann seedlings (Figure 4A,E). Therefore, the reduced level of abundance of both the miR396 and *GRF7* transcripts in mannitol-stressed *drb1* seedlings is likely indicating that miR396-directed regulation of *GRF7* expression is defective in the absence of the functional activity of DRB1 in this mutant background in the presence of the osmotic stressing agent mannitol.

In *Arabidopsis*, *PHO2* forms the single target gene of miRNA-directed expression regulation by all six members of the *MIR399* gene family [43,44,45]. In response to mannitol stress, *PHO2* expression was significantly reduced by 4.5- and 6.7-fold in Col-0/Mann and *drb4*/Mann seedlings and elevated by 4.3- and 1.5-fold in *drb1*/Mann and *drb2*/Mann seedlings (Figure 5E). In mannitol-stressed Col-0, *drb1* and *drb4* seedlings, the documented expression trends for *PHO2* were expected, as opposing miR399 accumulation trends were obtained by sRNA-Seq and RT-qPCR analysis of these three *Arabidopsis* lines (Figure 4A,F). The sRNA-Seq analysis of *drb1*/Ns and *drb2*/Ns seedlings (Appendix A) strongly inferred that in addition to DRB1, DRB2 also plays a major role in regulating the production of most members of the *Arabidopsis MIR399* gene family. Therefore, reduced miR399 and *PHO2* transcript abundance in the *drb2*/Mann sample could potentially indicate that the targeting miRNA is scaling in abundance to its target transcript if the miR399 sRNA is directing the alternate mechanism of miRNA-directed expression regulation, translation repression, to add an additional layer of regulatory complexity to control *PHO2* transcript abundance, and hence PHO2 protein levels, in response to mannitol stress. RT-qPCR assessment of *LACCASE3* (*LAC3*) expression showed that the level of abundance of this miR408 target gene was significantly elevated by 13.5- and 2.5-fold in Col-0/Mann and *drb1*/Mann seedlings and mildly increased by 1.4- and 1.1-fold in *drb2*/Mann and *drb4*/Mann seedlings (Figure 5F). The abundance of the targeting miRNA, miR408, was revealed by both sRNA-Seq and RT-qPCR to be significantly elevated in the Col-0/Mann and *drb1*/Mann samples (Figure 4A,G). This finding suggests that although miR408 abundance was elevated in these two *Arabidopsis* lines post the application of mannitol stress, the increase in miR408 accumulation did not occur to an adequate degree to compensate for the much greater increase to the level of expression of the *LAC3* target gene. RT-qPCR also showed that miR408 abundance was reduced by 1.1- and 1.4-fold in *drb2*/Mann and *drb4*/Mann seedlings (Figure 4G), respectively, an alteration in miR408 abundance which would account for the mildly elevated level of *LAC3* target gene expression observed in these two lines (Figure 5F) if target transcript cleavage formed the predominant mode of miR408-directed expression regulation to control *LAC3* transcript levels in *Arabidopsis* seedlings. Figure 5G shows that the expression of the assessed miR858 target gene, *ETHYLENE RESPONSE FACTOR7* (*ERF7*), which encodes a member of the *ERF* clade of the *ERF*/*APETELLA2* gene family of transcription factors [54], was mildly reduced in Col-0, *drb1*, *drb2* and *drb4* seedlings following the exposure of the four assessed *Arabidopsis* lines to the 7-day mannitol stress period. However, a highly elevated level of *ERF7* expression was expected in all four *Arabidopsis* lines following mannitol stress, considering that RT-qPCR (Figure 4H) indicated the abundance of miR858 to be significantly reduced by 10.0-, 3.3-, -3.8- and 2.7-fold in Col-0/Mann, *drb1*/Mann, *drb2*/Mann and *drb4*/Mann seedlings, respectively. Therefore, when considered together, the two sets of RT-qPCR analyses indicate that in 15-day-old *Arabidopsis* seedlings, miR858 exerts little regulatory effect over the expression of its *ERF7* target gene in response to the imposed stress.

## 3. Discussion

### 3.1. Assessment of the Phenotypic and Physiological Consequences of a 7-Day Mannitol Stress Treatment Regime on 15-Day-Old Col-0, drb1, drb2 and drb4 Seedlings

The primary aim of this study was to determine whether the loss of function of DRB1, DRB2 or DRB4 had the greatest impact on the ability of *Arabidopsis* seedlings to mount an effective miRNA-mediated molecular response to a 7-day growth period in the presence of the osmotic stress agent mannitol. The assessment of the phenotypic properties of whole seedling fresh weight, rosette area, and primary root length, all form routinely characterized metrics to ascertain the degree of sensitivity or tolerance of individual *Arabidopsis* lines to abiotic stress [55,56,57]. Quantification of these three phenotypic metrics for *drb1*, *drb2* and *drb4* seedlings failed to definitively establish which of the three assessed mutant backgrounds was the most sensitive or tolerant to the imposed stress (Figure 1B–D). However, this initial set of analyses readily showed that the phenotypic response of *drb1*, *drb2* and *drb4* seedlings differed considerably to that of Col-0 seedlings. Using the comparison of the phenotypic metrics of rosette area and primary root length for Col-0/Mann and *drb2*/Mann seedlings as an example, the rosette area of mannitol-stressed Col-0 seedlings was reduced by 61.7%, whereas the rosette area of *drb2*/Mann seedlings was reduced by a greater degree, 74.7% (Figure 1C). Similarly, the primary root length of Col-0/Mann seedlings was mildly promoted by 4.8%, whereas the primary root length of *drb2* seedlings was significantly reduced by 17.3% in response to the imposed stress (Figure 1D). Therefore, when considered together, the phenotypic assessments presented in Figure 1B–D infer that the ability of the *drb1*, *drb2* and *drb4* mutant backgrounds to mount an appropriate miRNA-mediated phenotypic response to this form of osmotic stress was rendered defective in the absence of the functional activity of DRB1, DRB2 and DRB4.

The abundance of anthocyanin is increased in *Arabidopsis* post its exposure to a range of abiotic stresses due to the ability of this flavonoid pigment to scavenge the toxic biomolecules, namely reactive oxygen species (ROS), produced by the exposure of *Arabidopsis* to abiotic stress [58,59,60]. In response to mannitol stress, anthocyanin content was only mildly to moderately altered in Col-0 (−0.5%), *drb1* (+23.6%), *drb2* (−3.5%) and *drb4* (+8.6%) seedlings (Figure 1E), which, when taken together, suggested than the 7-day cultivation period on solid growth medium containing 200 mM mannitol did not generate a level of ROS in 15-day-old *Arabidopsis* seedlings that promoted the accumulation of anthocyanin to any great extent. However, it is important to note here that the content of anthocyanin was increased by greater than 20-fold in the *drb1*/Mann sample, compared to the Col-0/Mann sample (Figure 1E). Considering that (1) DRB1 functions with DCL1 to produce most *Arabidopsis* miRNAs [14,15,16,21,24,25], and (2) miRNAs such as miR160, miR165/166, miR173, miR398 and miR408 have been associated with either ROS production or scavenging in *Arabidopsis* [61,62,63], the much greater alteration to anthocyanin abundance in mannitol-stressed *drb1* seedlings, compared to Col-0/Mann seedlings, may indicate that miRNA-directed regulation of the ROS biosynthesis pathways or of ROS scavenging mechanisms is defective in the absence of DRB1 activity. In contrast to anthocyanin abundance, the content of the primary photosynthetic pigment Chl *a* [63] and the important auxiliary photosynthetic pigment Chl *b* [64] were significantly reduced in Col-0, *drb1*, *drb2* and *drb4* seedlings by the imposed stress (Figure 1F,G). Interestingly, compared to the Col-0/Mann sample, the abundance of both assessed photosynthetic pigments was reduced to a much greater degree in mannitol-stressed *drb2* seedlings. More specifically, Chl *a* and *b* content was reduced by 23.8% and 23.4% in Col-0/Mann seedlings, compared to the more than 2-fold greater reductions of 57.7% and 58.1% determined for *drb2*/Mann seedlings (Figure 1F,G). This finding suggests that miRNA-directed regulation of components of the photosynthesis pathway was rendered defective in the *drb2* mutant background by mannitol-induced osmotic stress. In support of this, via analysis of the proteomes of the *drb1* and *drb2* single mutants, we have previously shown that a large and distinct cohort (i.e., distinct to the cohort of proteins with altered abundance in the *drb1* mutant) of 23 proteins assigned roles to the ‘chlorophyll biosynthesis process’ were reduced in abundance in the *drb2* mutant background [29]. Reduced abundance of this protein cohort may explain the greater reduction of Chl *a* and *b* content in *drb2*/Mann seedlings, compared to the level of reduction documented for Col-0/Mann seedlings. This distinct protein cohort involved in chlorophyll biosynthesis identified in the *drb2* mutant background [29] may almost account for the difference in the degree of reduction of Chl *a* and *b* content in mannitol-stressed *drb2* seedlings, compared to the level of Chl *a* and *b* content reduction in *drb1*/Mann and *drb4*/Mann seedlings (Figure 1F,G). In summary, when each of the phenotypic and physiological analyses conducted in this study are considered together (Figure 1), the greatest degree of reduction to rosette area, Chl *a* and Chl *b* content was observed in *drb2*/Mann seedlings, findings which suggest that of the three *drb* mutants analyzed, the *drb2* mutant was the most sensitive to mannitol-induced osmotic stress.

### 3.2. Profiling of the miRNA Landscapes of 15-Day-Old Mannitol-Stressed Col-0, drb1, drb2 and drb4 Seedlings

The phenotypic and physiological characterization of 15-day-old Col-0, *drb1*, *drb2* and *drb4* seedlings following a 7-day treatment period with the osmotic stress agent mannitol clearly showed that the imposed stress negatively impacted the developmental progression of each of the four assessed *Arabidopsis* lines to different degrees (Figure 1). Furthermore, the induction of a ‘stress response’ was confirmed for Col-0/Mann, *drb1*/Mann, *drb2*/Mann and *drb4*/Mann seedlings via molecular profiling of the expression (Figure 2A) of the well-documented stress response gene, *P5CS1* [33,34]. It is important to note here that the expression of *P5CS1* was induced to a greater degree in mannitol-stressed *drb1*, *drb2* and *drb4* seedlings than it was in Col-0/Mann seedlings (Figure 2A). Furthermore, *P5CS1* expression was induced to a higher level in *drb1*/Mann seedlings (+8.0-fold) than in either *drb2*/Mann (+2.4-fold) or *drb4*/Mann (+2.9-fold) seedlings. A higher level of *P5CS1* expression induction in the three assessed *drb* mutant backgrounds following the imposed stress treatment regime may indicate that a fully functional miRNA pathway is required for 15-day-old *Arabidopsis* seedlings to mount an appropriate molecular response to mannitol-induced osmotic stress. This suggestion is further supported by the higher degree of *P5CS1* expression induction in the *drb1* mutant background than in either the *drb2* or *drb4* mutant backgrounds, with DRB1 shown to be required to function together with DCL1 to produce most of the miRNAs which accumulate in *Arabidopsis* tissues [14,15,16,21,24,25].

The difference to which *P5CS1* expression was altered in mannitol-stressed Col-0, *drb1*, *drb2* and *drb4* plants suggested that the miRNA-mediated molecular response to the imposed stress would also differ across the four assessed *Arabidopsis* lines, and indeed, *Arabidopsis* line-specific alterations to the miRNA landscapes of mannitol-stressed Col-0, *drb1*, *drb2* and *drb4* seedlings were observed (Figure 3). More specifically, the abundance of 46.9% (n = 123/262) and 45.2% (n = 100/221) of the total number of miRNAs detected in Col-0 and *drb1* seedlings was significantly altered by the imposed stress. In contrast, only 22.5% (n = 58/258) and 32.2% (n = 95/295) of the total number of miRNAs detected in *drb2* and *drb4* seedlings had significantly altered abundance post the mannitol-induced osmotic stress. This result suggested that of the three *drb* mutants assessed, the ability to mount an appropriate miRNA-mediated molecular response to the imposed stress was most severely compromised in the *drb2* mutant. Interestingly, the phenotypic and physiological analyses presented in Figure 1 also indicated that *drb2* seedlings were the most sensitive to mannitol stress, findings which may be the direct result of the inability of the *drb2* mutant to mount an effective miRNA-mediated molecular response to this form of osmotic stress. The Figure 3 heatmap also shows the general trend of elevated accumulation for miRNAs with significantly altered abundance in Col-0/Mann (n = 119/123; 96.7%) and *drb4*/Mann (n = 86/95; 90.5%) seedlings, versus the general trend of reduced accumulation for miRNAs with significantly altered abundance in *drb1*/Mann and *drb2*/Mann seedlings, where 94.0% (n = 94/100) and 70.7% (n = 41/58) of significantly altered miRNAs were reduced in their abundance, respectively, post the imposed stress. The opposing accumulation trends for *drb1*/Mann and *drb2*/Mann seedlings, compared to Col-0/Mann seedlings, readily shows the degree to which the ability of these two mutant backgrounds to mount an appropriate miRNA-mediated molecular response to mannitol-induced osmotic stress was compromised in the absence of the functional activity of DRB1 and DRB2.

### 3.3. The Functional Activity of DRB1, DRB2 and DRB4 Are Required for the Production of miR160, miR164, miR167, miR396, miR399, miR408 and miR858 in 15-Day-Old Arabidopsis Seedlings

Further evidence that the miRNA-mediated molecular response to the imposed stress treatment regime was likely to be specific to each assessed *Arabidopsis* line was provided by sRNA-Seq assessment of miRNA accumulation in 15-day-old control-grown *drb1*, *drb2* and *drb4* seedlings for comparison to the Col-0/Ns sample (Appendix A). Considering that DRB1 is the primary functional partner of DCL1 for the production of most *Arabidopsis* miRNAs [14,15,16,21,24,25], it was unsurprising to document an approximate two-fold or greater reduction to miR160, miR164, miR167 and miR396 abundance in *drb1*/Ns seedlings, compared to Col-0/Ns seedlings (Appendix A). Similarly, we have previously shown that DRB2 is antagonistic to the action of DRB1 in the DCL1/DRB1 functional partnership as part of the production of specific miRNA subsets in specific *Arabidopsis* tissues [22,23,26,28], a finding that we again demonstrate here via the mildly elevated abundance of miR160, miR164, miR167 and miR396 in *drb2*/Ns seedlings. However, considering that the involvement of DRB4, together with DCL4, in the production stage of the *Arabidopsis* miRNA pathway has been previously reported to be restricted to the non-conserved subclass of *Arabidopsis* miRNAs [23,24,25], it was unexpected to observe that the abundance of members of these four highly conserved *MIR* gene families was reduced in *drb4*/Ns seedlings (Appendix A). Pélissier et al. [23] have shown that the accumulation of 24 nucleotide (nt) small-interfering RNAs (siRNAs) derived from DNA inverted repeats that do not rely on the activity of DNA-dependent RNA polymerase IV (Pol IV) for their production are greatly increased in abundance in *drb4* floral and vegetative tissues. Therefore, instead of demonstrating a previously unknown requirement for DRB4, together with DCL4, to regulate the production of the miR160, miR164, miR167 and miR396 sRNAs from their precursor transcripts in 15-day-old *Arabidopsis* whole seedlings, reduced levels of these four highly conserved plant miRNAs in *drb4*/Ns seedlings is more likely to be the result of a substantial shift in the representative proportions of each sRNA species that contributes to the global sRNA population of the *drb4* mutant. Moreover, loss of DRB4 function would result in the Pol IV-independent inverted repeat-derived 24 nt siRNA class of sRNA accounting for a much greater proportion of the global sRNA population of the *drb4* mutant. Therefore, compared to the contribution that this sRNA class makes to the global sRNA population of Col-0/Ns seedlings, the representative proportions contributed by the other sRNA classes, which also continue to accumulate in the tissues of the *drb4* mutant, and therefore, which also contribute to the global sRNA population of the *drb4* mutant, would be substantially reduced.

Our previous detailed molecular characterization of the miR399/*PHO2* expression module revealed that in addition to the primary regulatory role mediated by DRB1, DRB2 and DRB4 also direct secondary roles in regulating miR399 precursor transcript processing to fine-tune the level of miR399 production from its precursor transcripts [65]. Via profiling miR399 abundance in *drb1*/Ns, *drb2*/Ns and *drb4*/Ns seedlings, here we confirm that DRB1 is the primary nucleus-localized DRB protein involved in miR399 production, and that DRB2 and DRB4 both mediate secondary roles in the production of this miRNA in *Arabidopsis* seedlings (Appendix A). This finding not only demonstrates the regulatory complexity involved in miR399 production, but also reveals the importance of fine tuning the abundance of the *PHO2* transcript, and therefore the PHO2 protein, to ensure that Pi is appropriately distributed throughout *Arabidopsis* tissues during normal development and when *Arabidopsis* is exposed to environmental stress. Similarly, elevated miR408 abundance in *drb1*/Ns, *drb2*/Ns and *drb4*/Ns seedlings indicates that all three nucleus-localized DRB proteins mediate regulatory roles in modulating the rate of production of this miRNA in *Arabidopsis* seedlings (Appendix A). The highest level of accumulation elevation (+1.9-fold) for miR408 was detected in the *drb2*/Ns sample, a finding which strongly infers that DRB2 is highly antagonistic towards the action of both DRB1 and DRB4 in the DCL1/DRB1 and DCL4/DRB4 functional partnerships, as has been demonstrated previously [23,26,27,28]. However, considering that miR408 is a highly conserved miRNA in plants, elevated miR408 levels in *drb2*/Ns seedlings is most likely due to the removal of DRB2 antagonism of the DCL1/DRB1 functional partnership, the DCL/DRB partnership responsible for the production of most *Arabidopsis* miRNAs [15,16,21,24,25], and not antagonism of the DCL4/DRB1 functional partnership, demonstrated to be required for the production of only the non-conserved subclass of *Arabidopsis* miRNAs [22,23,25]. The miRNA, miR858, is classed as a non-conserved miRNA, and profiling of its level of accumulation by sRNA-Seq in non-stressed *drb1*, *drb2* and *drb4* seedlings confirmed that DRB4 is the primary nucleus-localized DRB protein required to produce miR858 in *Arabidopsis* via its functional interaction with DCL4 [22,23,25]. The profiling data did however, also show that DRB1 and DRB2 both mediate secondary roles in miR858 production (Appendix A), with the mildly elevated level of miR858 accumulation in the *drb1*/Ns and *drb2*/Ns samples indicating that their secondary role in regulating the production of this non-conserved miRNA most likely occurs via DRB1 and DRB2 binding to the miR858 precursor transcripts to fine-tune the interaction of DRB4 with the *PRE-MIR858A* and *PRE-MIR858B* precursors, an antagonistic mode of action which would enable tighter regulation of the rate of miR858 production by DCL4/DRB4.

### 3.4. The miRNA-Mediated Molecular Response of Arabidopsis Seedlings to Mannitol-Induced Osmotic Stress Is Defective in the Absence of DRB1, DRB2 and DRB4 Functional Activity

In addition to *ARF10* and *ARF16*, the *ARF17* transcript forms a target of miR160-directed expression regulation to contribute towards the molecular control of various aspects of *Arabidopsis* development [48,49]. Recently, via a molecular modification approach [66], we showed that in the root and shoot tissues of *Arabidopsis* seedlings, miR160 exclusively controls *ARF17* gene expression via a transcript cleavage mode of RNA silencing. However, in response to mannitol-induced osmotic stress, a miR160-directed transcript cleavage mode of RNA silencing was only observed in the *drb4*/Mann sample, with *ARF17* transcript abundance elevated by 1.8-fold (Figure 5A) in response to the 3.6-fold reduction in the level of miR160 (Figure 4B). In contrast, miR160 abundance scaled in accordance with the level of *ARF17* expression in *drb1*/Mann and *drb2*/Mann seedlings. Moreover, *ARF17* expression was reduced by 1.5- and 1.3-fold (Figure 5A) in response to 4.0- and 2.5-fold reductions to the level of the targeting miRNA, miR160, in mannitol-stressed *drb1* and *drb2* seedlings, respectively (Figure 4B). Scaling of the abundance of the regulating miRNA in accordance with the level of expression of its target transcript indicated that in *drb1*/Mann and *drb2*/Mann seedlings, miR160 was primarily controlling the level of *ARF17* expression by translational repression, the alternate mode of RNA silencing employed by *Arabidopsis* miRNAs [26,27,28]. Although miR164 has been shown to be responsive to various forms of osmotic stress in rice, wheat, potato and desert poplar [35,36,37,38], the 7-day cultivation period in the presence of 200 mM mannitol failed to influence the level of abundance of miR164 in Col-0, *drb1*, *drb2* and *drb4* seedlings to any great degree (Figure 4C). The expression of the assessed target gene of miR164, *CUC1* [50,51] was however, significantly reduced by 2.0-, 2.9- and 3.0-fold by the imposed stress in Col-0/Mann, *drb1*/Mann and *drb2*/Mann seedlings (Figure 5B). Significantly altered *CUC1* expression in the absence of change to miR164 abundance indicates that either (1) the ability of the miR164 sRNA to regulate *CUC1* transcript abundance was defective under the applied stress conditions, or (2) the promoter region of the *CUC1* gene harbors *cis*-regulatory elements, which allows the *CUC1* locus to transcriptionally respond to this form of osmotic stress, while the promoter regions of the encoding loci of the precursor transcripts of miR164 in *Arabidopsis*, including the *MIR164A*, *MIR164B* and *MIR164C* genes, do not harbor such *cis*-regulatory elements, which would render these three *MIR164* genes incapable of also responding at the transcriptional level to this form of osmotic stress.

In *Arabidopsis*, miR167 has been shown to regulate *ARF6* and *ARF8* expression in various aspects of development, including the development of the *Arabidopsis* root system and flower formation [51,52] and furthermore, in rice, barrel clover and *Arabidopsis*, miR167 has been shown to be responsive to osmotic stress [35,39,40,41]. In mannitol-stressed Col-0, *drb1* and *drb2* seedlings, *ARF8* expression was significantly repressed by 2.5-, 3.8- and 2.5-fold, respectively, whereas in the *drb4*/Mann sample, *ARF8* transcript abundance was significantly elevated by 2.2-fold (Figure 5C). In all four assessed *Arabidopsis* lines, miR167 accumulation showed an opposing response to the imposed stress, with miR167 levels increased by 1.8-, 2.1- and 2.0-fold in Col-0/Mann, *drb1*/Mann and *drb2*/Mann seedlings, respectively, and reduced by 2.0-fold in the *drb4*/Mann sample (Figure 4D). The opposing trends documented for miR167 and its *ARF8* target transcript in mannitol-stressed Col-0, *drb1*, *drb2* and *drb4* seedlings readily demonstrated that miR167 was regulating *ARF8* gene expression via the canonical mode of RNA silencing directed by *Arabidopsis* miRNAs, target transcript cleavage [18,19,20], as part of the miRNA-mediated molecular response to osmotic stress. In mannitol-stressed *drb4* seedlings, RT-qPCR revealed *DRB1* expression to be significantly elevated by 2.9-fold (Figure 2C) and that *DRB2* expression was decreased by 2.6-fold (Figure 2D), *DRB1* and *DRB2* expression trends which were unique to the *drb4*/Mann sample. Decreased *DRB2* transcript abundance would remove DRB2 antagonism of DRB1 action in the DCL1/DRB1 functional partnership, with elevated *DRB1* expression leading to even further promotion of DRB1 action in this protein partnership central to miRNA production in *Arabidopsis*. This alteration potentially accounts for the opposite trends in transcript abundance of the components of the miR167/*ARF8* expression module in *drb4*/Mann seedlings, compared to those obtained for this miRNA/target gene expression module in Col-0/Mann, *drb1*/Mann and *drb2*/Mann seedlings.

The GRF7 transcription factor has been previously proposed to orchestrate a cascade of transcriptional changes crucial to the molecular-mediated adaptive response to mannitol stress [42]. In mannitol-stressed Col-0, *drb1*, *drb2* and *drb4* seedlings, *GRF7* expression was reduced by 2.7-, 1.8-, 2.0- and 1.7-fold, respectively (Figure 5D), a transcriptional response shared by the four assessed *Arabidopsis* lines, which strongly supports the requirement of altered *GRF7* expression, and therefore GRF7 protein abundance, as part of the molecular-mediated adaptive response of *Arabidopsis* to mannitol stress [42]. The 3.1-, 1.5- and 2.5-fold induction to miR396 accumulation in Col-0/Mann, *drb2*/Mann and *drb4*/Mann seedlings, respectively (Figure 4E), indicates that in these three *Arabidopsis* lines, miR396-directed *GRF7* transcript cleavage forms the predominant mode of RNA silencing directed by the miR396 sRNA to control the expression level of its *GRF7* target gene. In the *drb1*/Mann sample, miR396 abundance was significantly reduced by 4.2-fold (Figure 4E), which, when considered together with the documented reduction to *GRF7* expression in this sample (Figure 5D), indicates that in the absence of the functional activity of DRB1, miR396-directed regulation of *GRF7* expression via the target transcript cleavage mode of RNA silencing is rendered defective.

The expression of *PHO2* was significantly reduced by 4.5- and 6.7-fold in Col-0/Mann and *drb4*/Mann plants (Figure 5E) in response to the respective 2.7- and 1.7-fold increase to the abundance of miR399 in these two *Arabidopsis* lines (Figure 4F). Reduced *PHO2*/PHO2 abundance in mannitol-stressed Col-0 and *drb4* seedlings would remove the PHO2-mediated degradation of the phosphate transporter proteins that control Pi allocation in *Arabidopsis* [43,44,45,46,65], a finding that suggests that Pi transport was promoted during the imposed stress regime to ensure that this precious resource continues to be appropriately translocated between the root and shoot tissues of Col-0/Mann and *drb4*/Mann seedlings for use in cellular processes essential to the adaptive response of these two *Arabidopsis* lines to mannitol-induced osmotic stress. In contrast to the *PHO2* expression trends documented for the Col-0/Mann and *drb4*/Mann samples, *PHO2* expression was elevated by 4.3- and 1.5-fold in *drb1*/Mann and *drb2*/Mann seedlings, respectively (Figure 5E). Furthermore, the level of miR399 was revealed by RT-qPCR to be largely unchanged and elevated in mannitol-stressed *drb1* and *drb2* seedlings, respectively (Figure 4F), miRNA and target gene expression trends which indicated that miR399 regulation of the expression of its *PHO2* target gene via the transcript cleavage mechanism of RNA silencing was rendered defective in the absence of the functional activity of DRB1 and DRB2. We have previously reported the requirement of DRB1 and DRB2 for miR399-directed *PHO2* expression regulation, and that disruption of the miR399/*PHO2* expression module in these two *drb* mutant backgrounds results in defective Pi allocation between the root systems and aerial tissues of the *drb1* and *drb2* mutants [65].

*LAC3* transcript abundance was elevated 13.5-, 2.5-, 1.4- and 1.1-fold in mannitol stressed Col-0, *drb1*, *drb2* and *drb4* seedlings, respectively (Figure 5F), with this shared expression trend indicating that elevated levels of *LAC3*/LAC3 form an essential molecular requirement in *Arabidopsis* at the seedling stage of development to respond to mannitol-induced osmotic stress. Indeed, the requirement of elevated *LAC3*/LAC3 levels in *Arabidopsis* seedlings has been demonstrated previously by Ma et al., [8] who showed that the *mir408* mutant line with elevated target gene expression was more resistant to PEG-induced osmotic stress than were wild-type *Arabidopsis* seedlings. Furthermore, in the same study [8], the molecular characterization of an additional *Arabidopsis* line that had been molecularly modified to over-accumulate miR408 and that had lower levels of miR408 target gene expression, including reduced *LAC3* gene expression, was shown to be more sensitive to PEG-induced osmotic stress than were unmodified wild-type *Arabidopsis* seedlings. Therefore, when these previous findings [8] are considered together with the *LAC3* expression data presented here in Figure 5F, elevated *LAC3*/LAC3 abundance appears to form an absolute requirement as part of the molecular response of *Arabidopsis* to osmotic stress. Similarly, repression of *ERF7* expression appears to also form a required molecular response to mannitol-induced osmotic stress in *Arabidopsis* seedlings, with *ERF7* transcript abundance moderately reduced in all four assessed *Arabidopsis* lines following the 7-day mannitol stress treatment period (Figure 5G). It is important to note here that the abundance of the *ERF7* targeting miRNA, miR858, was also reduced in Col-0/Mann, *drb1*/Mann, *drb2*/Mann and *drb4*/Mann seedlings (Figure 4H). This shared trend in transcript abundance for the miR408 sRNA and its *ERF7* target gene could either indicate that (1) translational repression forms the predominant mode of RNA silencing directed by miR858 to control the level of expression of its *ERF7* target gene, or (2) the promoter regions of the miR858 encoding loci, *MIR858A* and *MIR858B*, and of the *ERF7* target gene harbor the same complement of *cis*-regulatory elements, which would direct the same transcriptional response of all three loci to the imposed stress.

## 4. Materials and Methods

### 4.1. Plant Lines and Growth Conditions

The *Arabidopsis* single knockout mutants *drb1*, *drb2* and *drb4*, which harbor T-DNA insertions in the coding sequences of the *DRB1* (*AT1G09700*), *DRB2* (*AT2G28380*) and *DRB4* (*AT3G62800*) genes, respectively, have been described previously [22,23,26]. Col-0, *drb1*, *drb2* and *drb4* seeds were surface sterilized using chlorine gas for 90 min in a sealed chamber at room temperature, and post sterilization, seeds were transferred onto culture plates containing solid growth medium (half-strength Murashige and Skoog (MS) salts). The plates were sealed with gas-permeable tape and the seeds stratified via a 48 h (h) incubation period at 4 °C in the dark. The plates were then transferred to a temperature-controlled growth cabinet (A1000 Growth Chamber, Conviron^®^, Melbourne, Australia) and the seeds cultivated for 8 days, with a light/dark cycle of 16 h/8 h and a day/night temperature of 22 °C/18 °C. Equal numbers of 8-day-old Col-0, *drb1*, *drb2* and *drb4* seedlings (n = 48; 4 × plates with 12 seedlings per plate) were transferred to either (1) fresh plates containing standard growth medium (control plants (Ns plants)) or plates containing growth medium supplemented with 200 millimolar (mM) mannitol (mannitol-stressed plants (Mann plants)). Post seedling transfer, the newly prepared plates were again sealed with gas-permeable tape and returned to a temperature-controlled growth cabinet for a further 7-day period of cultivation under the same growth conditions as outlined above.

### 4.2. Phenotypic and Physiological Assessments

At day 15, control or mannitol-stressed Col-0, *drb1*, *drb2* and *drb4* seedlings were sampled for whole-seedling fresh weight determination to assess the consequence of the imposed stress on *Arabidopsis* development. For the determination of rosette area (millimeters squared (mm^2^)) and primary root length (mm), photographic images of 15-day-old control or mannitol-stressed Col-0, *drb1*, *drb2* and *drb4* seedlings were analyzed via the use of the online software ImageJ (https://imagej.net/ij/). It is important to note here that for the assessment of primary root length, 8-day-old Col-0, *drb1*, *drb2* and *drb4* seedlings grown under standard *Arabidopsis* growth conditions were transferred to freshly prepared control or mannitol stress plates exactly as outlined above. However, post seedling transfer and sealing of the plates with gas-permeable tape, the plates were orientated vertically in the temperature-controlled growth cabinet for the 7-day assessment period.

One hundred milligrams (mg) of freshly sampled control and mannitol-stressed Col-0, *drb1*, *drb2* and *drb4* whole seedlings were ground into a fine powder using liquid nitrogen (LN_2_), with each powdered sample immediately transferred to 1.0 milliliter (mL) of acidic methanol (99 mL of 100% methanol and 1.0 mL of concentrated hydrochloric acid) to determine the anthocyanin content of the samples via standard spectrophotometry. Samples were incubated for 2 h at 4 °C and then centrifuged for 5 min at 15,000× *g* at room temperature. The resulting supernatants were immediately transferred to individual cuvettes, and the absorbance of each sample was determined using a GENESYS 10S spectrophotometer (Thermo Fisher Scientific, Sydney, Australia) at wavelengths of 530 and 657 nanometers (nm; A_530_ and A_657_) and via the use of acidic methanol as the reagent blank. Post this analysis, the anthocyanin content of control and mannitol-stressed Col-0, *drb1*, *drb2* and *drb4* whole seedlings was determined, as outlined in [67], via the use of the equation: anthocyanin content (micrograms/gram fresh weight (μg/g FW)) = A_530_ − 0.25 × A_657_/sample weight (g).

To determine the chlorophyll *a* and *b* content of control and mannitol-stressed Col-0, *drb1*, *drb2* and *drb4* whole seedlings, 100 mg of freshly sampled material was ground into a fine powder under LN_2_. One milliliter of 80% (*v*/*v*) acetone was added to each sample, and samples were incubated at room temperature for 24 h in the dark, and then centrifuged at 15,000× *g* for 5 min at room temperature. The resulting supernatants were transferred to individual cuvettes, and the absorbance of each sample was assessed in a GENESYS 10S spectrophotometer (Thermo Fisher Scientific, Sydney, Australia) at wavelengths 646 and 663 nm and via the use of 80% (*v*/*v*) acetone as the reagent blank. The A_646_ and A_663_ values were then used to determine the chlorophyll *a* and *b* content of control and mannitol-stressed Col-0, *drb1*, *drb2* and *drb4* whole seedlings via applying the obtained values to the Lichtenthaler’s equations as outlined in [68]. These initially determined values were then converted to final value of micrograms per gram of fresh weight (μg/g FW).

### 4.3. Nucleic Acid Preparations and Quantification of miRNA Abundance and Target Gene Expression via RT-qPCR

Total RNA was extracted from four biological replicates, with each replicate consisting of 100 mg of pooled freshly sampled control and mannitol-stressed Col-0, *drb1*, *drb2* and *drb4* whole seedlings using TRIzol™ Reagent according to the protocol of the manufacturer (Thermo Fisher Scientific, Sydney, Australia). It is important to note here that a second set of four biological replicates, each consisting of a pool of 12 individual plants, was used for the molecular analyses performed in this study due to the destructive nature of the physiological experiments. A NanoDrop^®^ ND-1000 spectrophotometer (Thermo Fisher Scientific, Sydney, Australia) was used to determine the total RNA concentration (μg/μL) of each preparation, and the quality of each extraction was confirmed via the use of standard electrophoretic separation of nucleic acids on an ethidium bromide-stained 1.2% (*w*/*v*) agarose gel.

For each high-quality total RNA extraction, 5.0 μg of total RNA was treated with 5.0 units (U) of DNase I to digest any contaminating genomic DNA according to the manufacturer’s protocol (New England Biolabs, Melbourne, Australia), and post DNase I treatment, each sample was further purified via the use of a RNeasy Mini kit (Qiagen, Melbourne, Australia). One microgram (1.0 μg) of purified total RNA was used as the template to synthesize a standard high molecular weight complementary DNA (cDNA) library via the use of 1.0 U of ProtoScript^®^ II Reverse Transcriptase along with 2.5 mM of oligo dT_(18)_ according to the manufacturer’s protocol (New England Biolabs, Melbourne, Australia). Five hundred nanograms (500 ng) of purified total RNA was used as the template to generate miRNA-specific cDNAs using miRNA-specific stem-loop DNA oligonucleotides (Appendix A) and 1.0 U of ProtoScript^®^ II Reverse Transcriptase. For the synthesis of each miRNA-specific cDNA, the cycling conditions of (1) 1 cycle of 16 °C for 30 min, (2) 60 cycles of 30 °C for 30 s (s), 42 °C for 30 s and 50 °C for 2 s, and (3) 1 cycle of 85 °C for 5 min were used.

Post the synthesis of all cDNAs, each single-stranded cDNA was diluted to a working concentration of 50 ng/µL in RNase-free water prior to its use as a template for the quantification of the abundance of either a specific miRNA or to determine the level of gene expression. All RT-qPCR assessments of transcript abundance used the same cycling conditions of (1) 1 cycle of 95 °C for 10 min and (2) 45 cycles of 95 °C for 10 s and 60 °C for 15 s. The GoTaq^®^ qPCR Master Mix (Promega, Sydney, Australia) was used as the fluorescent reagent for all performed RT-qPCR experiments. miRNA abundance and gene expression were quantified using the 2^−∆∆CT^ method, with the small nucleolar RNA, snoR101, and *UBIQUITIN10* (*UBI10*; *AT4G05320*) used as the respective internal controls to normalize the relative abundance of each transcript under assessment. For all RT-qPCR experiments reported here, three technical replicates were performed for each of the four biological replicates of the control and mannitol-stressed Col-0, *drb1*, *drb2* and *drb4* samples. The sequence of each DNA oligonucleotide used in this study for either the synthesis of a miRNA-specific cDNA, or to quantify miRNA abundance or the expression level of a select miRNA target gene is provided in Appendix A.

### 4.4. Statistical Analysis of Phenotypic, Physiological and Molecular Data

Analytical data from this study were obtained from four biological replicates of the control and mannitol-stressed Col-0, *drb1*, *drb2* and *drb4* samples, and each biological replicate consisted of a pool of 12 individual plants. Statistical analysis was performed using a standard two-tailed *t*-test. The presence of an asterisk (*) above a column of a graph presented in Figure 1, Figure 2, Figure 4 and Figure 5 represents a statistically significant difference between the mannitol-stressed sample and its control-grown counterpart, with *p*-values: * < 0.05, ** < 0.005, and *** < 0.001.

## 5. Conclusions

This study assessed the phenotypic, physiological, and miRNA-mediated molecular response of 15-day-old Col-0, *drb1*, *drb2* and *drb4* seedlings to a 7-day cultivation period in the presence of the osmotic stress agent, mannitol. When taken together, the phenotypic and physiological analyses identified the *drb2* mutant as the most sensitive to the imposed stress, with whole-seedling fresh weight, Chl *a* and Chl *b* content all reduced to the greatest degree in mannitol-stressed *drb2* seedlings. At the molecular level, however, the opposing accumulation trends for the altered abundance miRNA populations of *drb1*/Mann and *drb2*/Mann seedlings, compared to those documented for Col-0/Mann and *drb4*/Mann seedlings, strongly inferred that the ability of *Arabidopsis* to mount an appropriate miRNA-mediated molecular response to mannitol stress was defective in the absence of the functional activity of either DRB1 or DRB2. Furthermore, detailed RT-qPCR assessment of the abundance trends of seven miRNAs in control-grown and mannitol-stressed *drb1*, *drb2* and *drb4* seedlings, including miR160, miR164, miR167, miR396, miR399, miR408 and miR858 modules, revealed the requirement of the hierarchical action of DRB1, DRB2 and DRB4 to fine-tune the rate of production of each of the seven assessed miRNAs. The RT-qPCR-based molecular assessments also allowed for the identification of the miRNA/target gene expression modules most likely to be crucial for *Arabidopsis* to mount an effective miRNA-mediated molecular response to mannitol-induced osmotic stress, including the miR396/*GRF7*, miR399/*PHO2* and miR408/*LAC3* expression modules. Furthermore, the identification of these three essential miRNA/target gene expression modules has identified ideal targets for molecular modification to generate novel *Arabidopsis* lines in the future with increased tolerance to this form of osmotic stress.

## Figures and Tables

**Figure 1 ijms-25-12562-f001:**
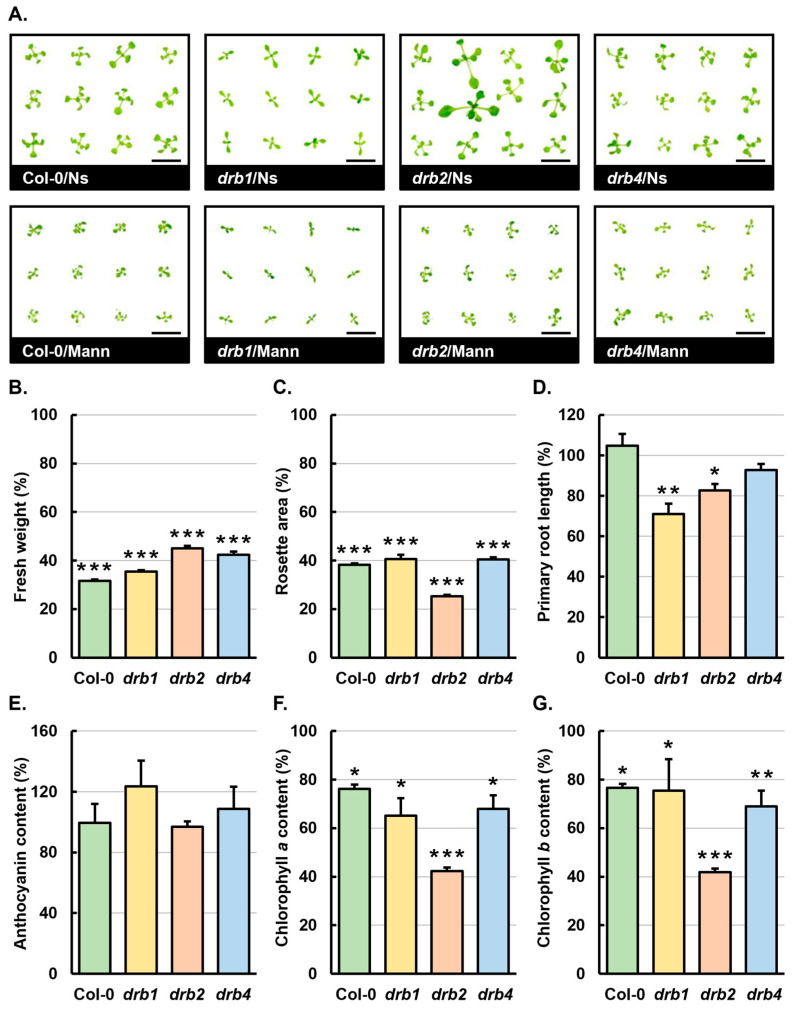
Phenotypic and physiological assessment of 15-day-old Col-0, *drb1*, *drb2* and *drb4* seedlings following a 7-day growth period on solid growth medium supplemented with 200 mM mannitol. (**A**) Phenotypes expressed by 15-day-old control-grown Col-0/Ns, *drb1*/Ns, *drb2*/Ns and *drb4*/Ns seedlings (top panel) and mannitol-stressed Col-0/Mann, *drb1*/Mann, *drb2*/Mann and *drb4*/Mann seedlings (bottom panels). Scale bar = 1.0 cm. (**B**–**G**) All quantified phenotypic and physiological parameters are compared to those obtained for the control-grown counterpart of each *Arabidopsis* line, with differences in whole-seedling fresh weight (**B**), rosette area (**C**), primarily root length (**D**), anthocyanin content (**E**) and chlorophyll *a* (**F**) and *b* (**G**) content determined via the assessment of four biological replicates which consisted of pools of 12 individual plants. (**B**–**G**) Error bars represent the standard deviation (±SD) of the four biological replicates, and the presence of an asterisk (*) above a column represents a statistically significant difference between the mannitol-stressed sample and the control sample (*p*-value: * < 0.05; ** < 0.005; *** < 0.001).

**Figure 2 ijms-25-12562-f002:**
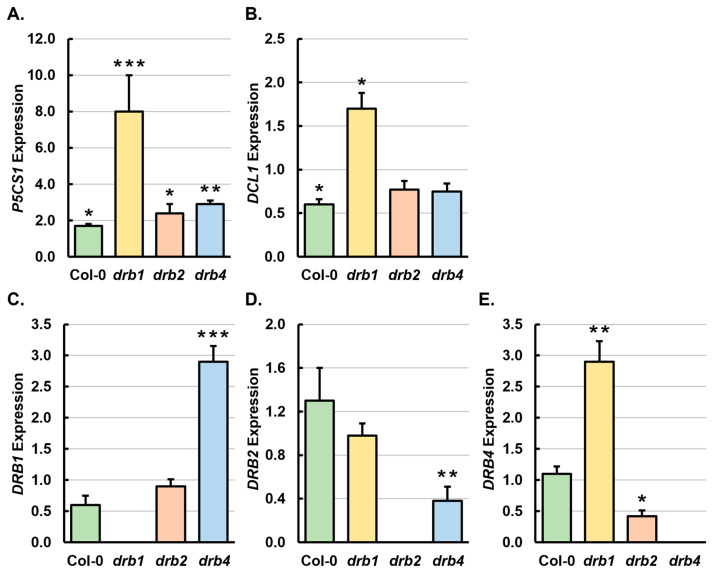
Molecular assessment via RT-qPCR of the effects of the exposure of 15-day-old Col-0, *drb1*, *drb2* and *drb4* seedlings to the osmotic stress agent mannitol. (**A**) RT-qPCR analysis of the expression of the *P5CS1* stress response gene in 15-day-old Col-0, *drb1*, *drb2* and *drb4* seedlings following a 7-day exposure period to mannitol stress. (**B**–**E**) RT-qPCR assessment of *DCL1* (**B**), *DRB1* (**C**), *DRB2* (**D**) and *DRB4* (**E**) gene expression in 15-day-old Col-0, *drb1*, *drb2* and *drb4* seedlings following a 7-day exposure period to osmotic stress. Error bars represent the ±SD of four biological replicates, and the presence of an asterisk (*) above a column represents a statistically significant difference between the sample exposed to stress and the control sample (*p*-value: * < 0.05; ** < 0.005; *** < 0.001).

**Figure 3 ijms-25-12562-f003:**
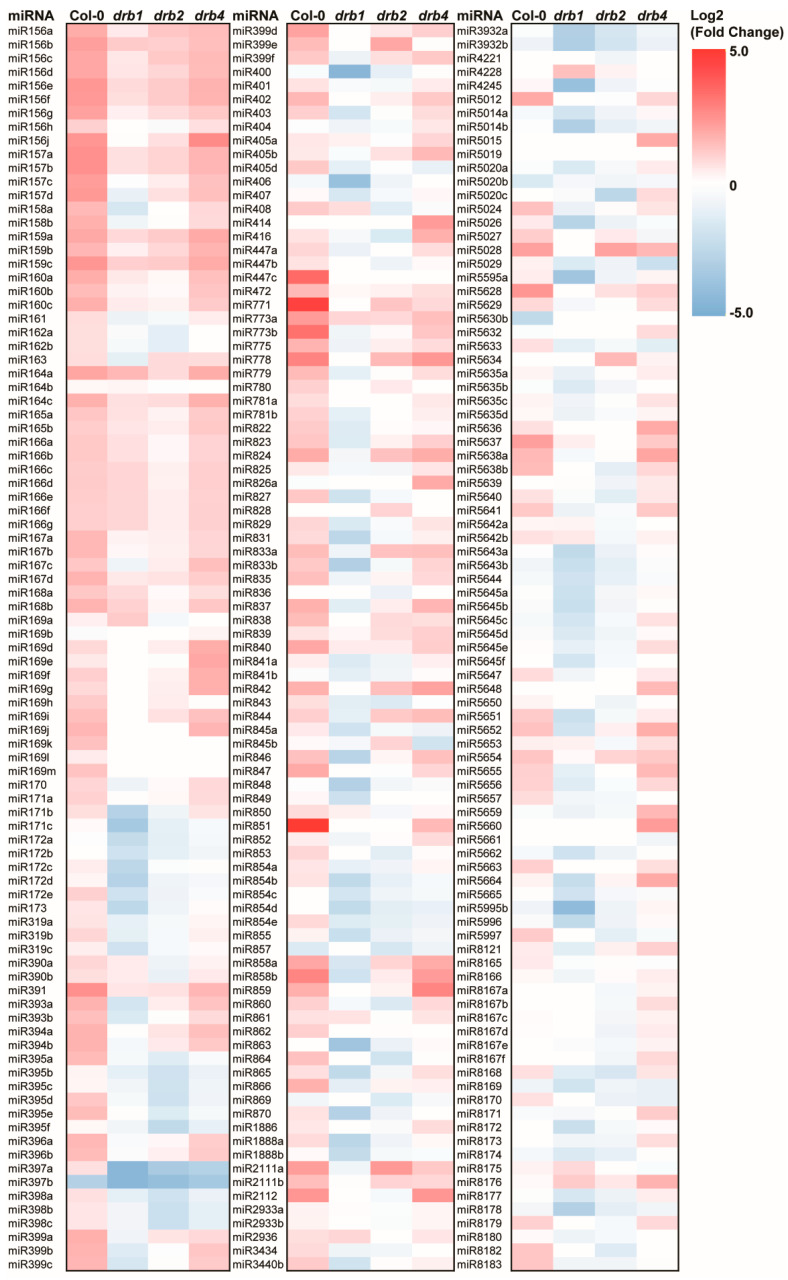
Profiling of the miRNA landscapes of 15-day-old mannitol stressed Col-0, *drb1*, *drb2* and *drb4* seedlings via small RNA sequencing. A sRNA-Seq approach was used to establish the degree of alteration to the miRNA landscapes of 15-day-old mannitol-stressed Col-0, *drb1*, *drb2* and *drb4* seedlings. For each vertical column of the heatmap, an individual tile represents a single miRNA per assessed *Arabidopsis* line, where the intensity of the red-colored shading indicates the extent of abundance upregulation for an individual miRNA, while the intensity of the blue-colored shading represents the extent of abundance downregulation for an individual miRNA.

**Figure 4 ijms-25-12562-f004:**
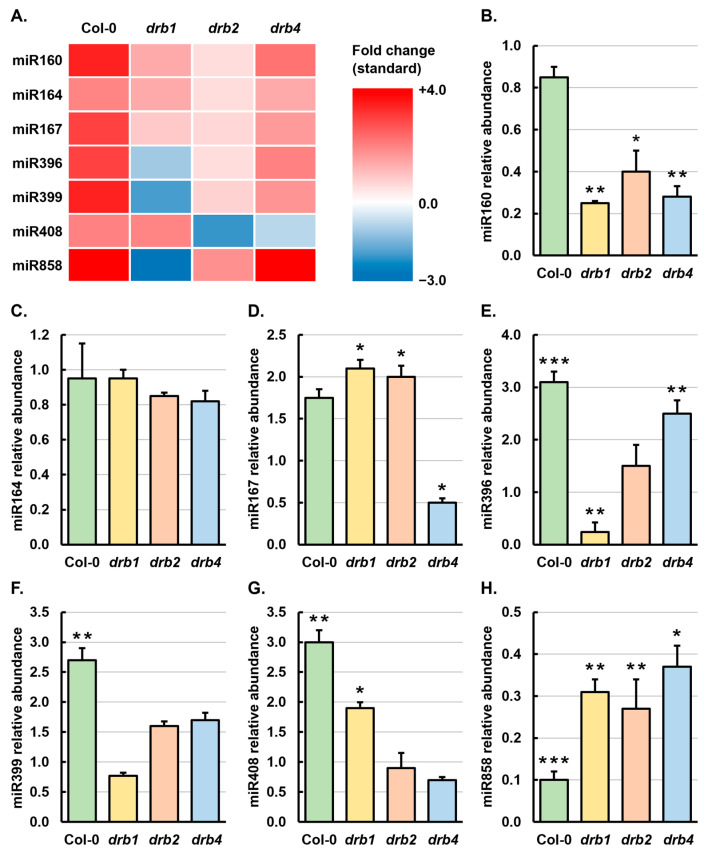
Profiling of miRNA accumulation in 15-day-old mannitol-stressed Col-0, *drb1*, *drb2* and *drb4* seedlings by sRNA-Seq and RT-qPCR. (**A**) Profiling by sRNA-Seq of the abundance of all members of the *MIR160*, *MIR164*, *MIR167*, *MIRR396*, *MIR399*, *MIR408* and *MIR858* gene families in mannitol-stressed Col-0, *drb1*, *drb2* and *drb4* seedlings. The shading intensity (light to dark) of each tile of each column depicts the degree of abundance change presented as a standard fold change. (**B**–**H**) RT-qPCR quantification of miR160 (**B**), miR164 (**C**), miR167 (**D**), miR396 (**E**), miR399 (**F**), miR408 (**G**) and miR858 (**H**) abundance in 15-day-old Col-0, *drb1*, *drb2* and *drb4* seedlings following the 7-day osmotic stress treatment period with miRNA abundance compared to the control grown counterpart of each *Arabidopsis* line. Error bars represent the ±SD of four biological replicates, and each biological replicate consisted of a pool of 12 individual plants. The presence of an asterisk (*) above a column represents a statistically significant difference between the mannitol-stressed sample and the control sample (*p*-value: * < 0.05; ** < 0.005; *** < 0.001).

**Figure 5 ijms-25-12562-f005:**
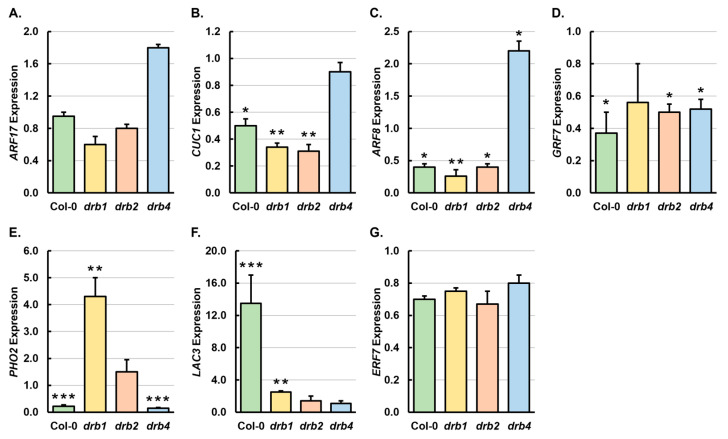
RT-qPCR assessment of miRNA target gene expression in 15-day-old mannitol-stressed Col-0, *drb1*, *drb2* and *drb4* seedlings. (**A**–**G**) RT-qPCR analysis of the expression of *ARF17* (**A**), *CUC1* (**B**), *ARF8* (**C**), *GRF7* (**D**), *PHO2* (**E**), *LAC3* (**F**) and *ERF7* (**G**), the representative target genes of miR160, miR164, miR167, miR396, miR399, miR408 and miR858, respectively, in 15-day-old mannitol-stressed Col-0, *drb1*, *drb2* and *drb4* seedlings. Target gene expression (presented as a standard fold change) in each mannitol-stressed *Arabidopsis* line was determined via direct comparison to the level of target gene expression in the control grown counterpart of each line. Error bars represent the ±SD of four biological replicates, with each biological replicate consisting of a pool of 12 individual plants. The presence of an asterisk (*) above a column represents a statistically significant difference between the mannitol-stressed sample and the control sample (*p*-value: * < 0.05; ** < 0.005; *** < 0.001).

## Data Availability

All data reported here are available from the authors upon request.

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
