# Peer review of "DRB1, DRB2 and DRB4 Are Required for an Appropriate miRNA-Mediated Molecular Response to Osmotic Stress in Arabidopsis thaliana"

_ijms, 2024, doi:10.3390/ijms252312562_

Round 1
Reviewer 1 Report
Comments and Suggestions for Authors
The manuscript "DRB1, DRB2 and DRB4 are required for an appropriate miRNA-mediated molecular response..." by Pegler et al. is a molecular study on how osmotic stress (mannitol) affects the expression of three DRB proteins, as well as how the abundance of miRNAs changes in Arabidopsis drb mutants.
The study follows a logical path, the methodology of which is meticulously described. The authors' discussion is supported by both new data presented here as well as appropriate literature. The case they built is well structured, and the paper lays out this logic in a proper way. Overall, I recommend publication of this interesting study.
Minor remarks:
- 2nd line on page 21: 220 : UNIT mM is MISSING
- par 4.4 p values equal: * < 0.05 ... ; also - is a multiple testing correction included in the Tukey test in SPSS (probably Benjamini Hochberg?)
- caption 4 last line : I think the 472 is a typo
- and then a very petty nitpick: in par. 2.3, the authors use the adverb clearly four times to refer to Fig. 3. While I do not dispute the clarity of the graph, I believe it can be cut at least twice.
Author Response
Response to Reviewer #1 comments:
The authors wish to take this opportunity to thoroughly thank Reviewer #1 for their highly positive review of our submitted manuscript. We also thank Reviewer #1 for identifying several oversights in our original submission. We address each of Reviewer #1’s specific comments below:
Comment 1: 2nd line on page 21: 220: UNIT mM is MISSING
Response 1: We thank Reviewer #1 for identifying this error. We have added ‘mM’ to the text of our revised manuscript as required.
Comment 2: par 4.4 p values equal: * < 0.05 ... ; also - is a multiple testing correction included in the Tukey test in SPSS (probably Benjamini Hochberg?)
Response 2: We have removed the word ‘equal’ from subsection 4.4. of the Materials and Methods section in the revised version of our manuscript. The authors thank Reviewer #1 for this helpful suggestion.
Furthermore, we have also extensively modified the text of subsection 4.4. of the Materials and Methods to more clearly outline to the reader the statistical analyses conducted in this study. Again, we thank Reviewer #1 for identifying this oversight. Subsection 4.4. text has been modified to “Analytical data from this study were obtained from four biological replicates of the control and mannitol-stressed Col-0, drb1, drb2 and drb4 samples, and each biological replicate consisted of a pool of 12 individual plants. Statistical analysis was performed using a standard two-tailed t-test. The presence of an asterisk (*) above a column of a graph presented in Figures 1, 2, 4 and 5 represents a statistically significant difference between the mannitol-stressed sample and its control grown counterpart with p-values: * < 0.05, ** < 0.005, and *** < 0.001.”
Comment 3: caption 4 last line : I think the 472 is a typo
Response 3: We thank Reviewer #1 for identifying this mistake in the Figure 4 legend. We have removed ‘472’ from the text of the Figure 4 legend as required.
Comment 4: and then a very petty nitpick: in par. 2.3, the authors use the adverb clearly four times to refer to Fig. 3. While I do not dispute the clarity of the graph, I believe it can be cut at least twice.
Response 4: The authors thank Reviewer #1 for bringing to our attention our overuse of the adverb ‘clearly’ as part of the text of subsection 2.3. of the Results. We have altered the text accordingly in subsection 2.3. of the Results section of our revised manuscript, so that ‘clearly’ is now only used once. Thank you again for identifying this oversight.
Reviewer 2 Report
Comments and Suggestions for Authors
How many individual plants were used in Figure 1? For experiments of this nature, at least 100 plants are needed to account for potential variations. Although fewer than 50 plants were mentioned in the Materials and Methods section, which seems to be a relatively low number for assessing phenotypic variation, experiments with such a small sample size may carry certain risks.
In Figure 2, it would be beneficial to show gene expression of DRB1, 2, and 4 in Col-0 under salt or mannitol stress, along with mutant backgrounds. Controls are, of course, crucial here; gene expression in the control (unstressed) condition should also be included. The same applies to Figure 5. While it is valuable to observe responses in 15-day-old plants subjected to mannitol stress, control conditions without stress must also be assessed.
The authors have identified genes regulated by miRNA, particularly selecting stress factors related to osmotic stress, and have studied gene expression for DRB1 and DRB2. Ideally, additional biochemical data should be provided to show that miRNA indeed binds to these genes or indicate the specific binding sites. Furthermore, evidence from transgenic lines with miRNA expression would strengthen the findings. Currently, the manuscript appears to conclude with only the identification of miRNA, which is insufficient. Additional research is needed to support these conclusions.
Author Response
Response to Reviewer #2 comments:
The authors wish to take this opportunity to thank Reviewer #2 for their time critically reviewing our submitted manuscript, your efforts are very much appreciated.
Below we address your specific comments stemming from your review:
Comment 1: How many individual plants were used in Figure 1? For experiments of this nature, at least 100 plants are needed to account for potential variations. Although fewer than 50 plants were mentioned in the Materials and Methods section, which seems to be a relatively low number for assessing phenotypic variation, experiments with such a small sample size may carry certain risks.
Response 1: The authors respectfully disagree with this comment raised by Reviewer #2. The number of individual plants used to generate the Figure 1 data = 4 x biological replicates with each biological replicate consisting of 12 individual plants = 48 plants per analysed sample. The number of individual plants used for this analysis is stated on numerous occasions throughout our manuscript. We do not understand where Reviewer #2 has obtained the number of ‘100 plants’ as an acceptable number for such analyses. Similar largescale studies performed in Arabidopsis and other plant species rely on much smaller sample sizes. Therefore, our use of 48 plants for the physiological analyses, and then the use of an additional 48 plants for the molecular assessments reported in our study is appropriate. Further, the use of 48 plants is appropriate for such experimentation in Arabidopsis as Arabidopsis is a self-pollinating species, and this, together with the very tightly controlled growth conditions used in this study removes considerable genetic, environmental and/or experimental variation. Finally, Reviewer #1 comments on the ‘meticulous’ nature of the experimentation conducted in this study. We, the authors, are therefore of the opinion that the number of plants used for our experimentation, and how our experimentation was performed in this study, is appropriate.
Comment 2: In Figure 2, it would be beneficial to show gene expression of DRB1, 2, and 4 in Col-0 under salt or mannitol stress, along with mutant backgrounds. Controls are, of course, crucial here; gene expression in the control (unstressed) condition should also be included. The same applies to Figure 5. While it is valuable to observe responses in 15-day-old plants subjected to mannitol stress, control conditions without stress must also be assessed.
Response 2: We, the authors, respectfully disagree with this suggestion by Reviewer #2. In this study we repeatedly compare control (non-stressed) versus stressed samples for each of the four individual Arabidopsis lines assessed. Via such an approach where we repeatedly compare control versus stressed samples we have incorporated the ‘controls’ directly into each of our analyses. Further, our study requires this repeated approach to Figure data generation/presentation to allow for simple comparison of the data presented in each manuscript Figure. Changing the way we present data, data which is already incorporated into our current series of Figures, in just manuscript Figures 2 and 5 would fail to allow for this direct comparison between the results presented in each of the manuscript’s Figures, and thus, lessening the impact of the data being presented.
In addition, we have published numerous studies previously outlining the differences in the level of expression of DRB1, DRB2 and DRB4 in wild-type Arabidopsis and in the drb mutant backgrounds. Therefore, repeating this analysis would provide little advancement on what is already known regarding the expressional and functional interplay between the individual DRB gene family members in Arabidopsis. The main aims of this study were to (1) identify mannitol-induced osmotic stress responsive miRNAs, and (2) determine the degree of involvement of DRB1, DRB2 and/or DRB4 in the production of the identified mannitol stress responsive miRNAs, or the mechanism of target gene expression regulation directed by either DRB1, DRB2 or DRB4. The authorship team is therefore of the opinion that the current format of data presentation in the manuscript’s Figures most clearly outlines these primary aims of our study.
Comment 3: The authors have identified genes regulated by miRNA, particularly selecting stress factors related to osmotic stress, and have studied gene expression for DRB1 and DRB2. Ideally, additional biochemical data should be provided to show that miRNA indeed binds to these genes or indicate the specific binding sites. Furthermore, evidence from transgenic lines with miRNA expression would strengthen the findings. Currently, the manuscript appears to conclude with only the identification of miRNA, which is insufficient. Additional research is needed to support these conclusions.
Response 3: Again, we the authors respectfully disagree with this Reviewer #2 comment. We, and others, have published studies previously outlining the requirement of DRB1, DRB2 and/or DRB4 for the production of specific miRNA subsets in Arabidopsis. We, and others, have also previously published works on what miRNAs are bound by the three assessed DRB proteins, as well as outlining specific binding sites for individual miRNAs and their targeted genes. Repeating such experimentation was not the aim of this study, and such work would not advance the current knowledge of the field. Our current study has identified very large numbers of miRNAs responsive to mannitol-induced osmotic stress, as well as to show that the global miRNA-mediated molecular response to this form of osmotic stress differs considerably in the absence of the functional activity of three key pieces of protein machinery known to be required for miRNA production and/or action in Arabidopsis. In addition, for the mannitol-stress responsive miRNAs experimentally validated, we also report on the mode of action directed by each miRNA to control target gene expression. Therefore, based on the considerable volume of novel findings presented in this study, the authors do not agree with the Reviewer #2 statement that ‘miRNA identification is insufficient’. Finally, the generation of transformant lines is well outside the scope of this present study, as such a research undertaking would form an entirely separate investigation in its own right.